



# Reconciling differences in stratospheric ozone composites

William T. Ball[1,2], Justin Alsing[3,4], Daniel J. Mortlock[4,5,6], Eugene V. Rozanov[1,2], Fiona Tummon[1], and Joanna D. Haigh[4,7]

[1]Institute for Atmospheric and Climate Science, Swiss Federal Institute of Technology Zurich, Universitaetstrasse 16, CHN, CH-8092 Zurich, Switzerland
[2]Physikalisch-Meteorologisches Observatorium Davos World Radiation Centre, Dorfstrasse 33, 7260 Davos Dorf, Switzerland
[3]Center for Computational Astrophysics, Flatiron Institute, 162 5th Ave, New York, NY 10010, USA
[4]Physics Department, Blackett Laboratory, Imperial College London, SW7 2AZ, UK
[5]Department of Mathematics, Imperial College London, SW7 2AZ, UK
[6]Department of Astronomy, Stockholms universitet, SE-106 91 Stockholm, Sweden
[7]Grantham Institute - Climate Change and the Environment, Imperial College London, SW7 2AZ, UK

*Correspondence to:* W. T. Ball (william.ball@env.ethz.ch)

**Abstract.** To accurately estimate decadal trends in stratospheric ozone requires stable long-term observations. Recently, several ozone composites have been published that combine observations from multiple instruments to span more than three decades. Despite this, trends disagree by latitude and altitude, even between composites built upon the same instrument data. We confirm that the leading

causes of differences in decadal trend estimates lie in (i) steps in the composite timeseries when the instrument source data changes and (ii) artificial sub-decadal trends in the underlying instrument data. These artefacts introduce features that can alias with regressors in multiple linear regression (MLR) analysis; both lead to inaccurate trend estimates. Here, we aim to remove these artefacts by applying particle filtering, sequential Monte Carlo Bayesian estimation, which uses only the data

itself in addition to prior knowledge about ozone variability and known problems during instrument operation. We apply the particle filter to stratospheric ozone in 10° bands from 60°S–60°N and from 46–1 hPa (~21–48 km) for 1985–2012. There are two main outcomes: (i) we independently identify and confirm many of the data problems previously identified, but which remain unaccounted for in existing composites; (ii) we construct an ozone composite, with uncertainties, that is free from most

of these problems. To analyse the new data series, we use dynamical linear modelling (DLM), which provides a more robust estimate of long-term changes through Bayesian inference than MLR. Particle filtering and DLM, together, provide a step forward in improving estimates of decadal trends. Our results indicate a significant recovery of ozone since 1998 in the upper stratosphere, of both northern and southern mid-latitudes, in all four composites analysed, and particularly in the new particle





filter composite. The particle filter results also show no hemispheric difference in the recovery at mid-latitudes, in contrast to a feature that is present, but not consistent, in the four composites. We recommend using the particle filter method to construct a new composite based not on existing composites, as we do here, but on the original instrument data: such a product would provide a further advance for the estimation of decadal changes in stratospheric ozone.

**1 Introduction**

The ozone layer in the stratosphere is vital for protecting the biosphere from harmful solar ultraviolet (UV) radiation. Damage to the ozone layer from the use of chloro-flurocarbons (CFCs) and other ozone depleting substances (ODSs) led to a decline in ozone globally over the latter half of the 20th Century (Johnston, 1971; Crutzen, 1971; Molina and Rowland, 1974), particularly in the polar

regions (WMO, 2011, 2014). The implementation of the Montreal Protocol (MP), which banned the use of most ODSs, has led to a halt of this decline and a slow recovery in total ozone has ensued in some regions (Solomon et al., 2016). However, there is low confidence in the sign and magnitude of recent trends depending on altitude and latitude, and a clear signal is difficult to determine (Harris et al., 2015).

Ozone responds to forcings from below, e.g. injections of aerosols from volcanoes (Robock, 2000) or wave activity from the troposphere (Kidston et al., 2015), and from above, e.g. from solar sources such as UV radiation (Haigh, 1994) and particles (Funke et al., 2011; Mironova et al., 2015). In order to quantify and understand the variability forced by a particular driver, and long-term trends in ozone - not just in terms of the total column ozone (TCO), but also resolved vertical profiles - observations

spanning multiple decades are needed. Such a dataset can only be provided by combining data from multiple sources (Tummon et al., 2015; Harris et al., 2015). The method used to combine the data needs to consider different inherent attributes, the most important of which include the: temporal resolution, vertical and horizontal spatial resolution (Kramarova et al., 2013a), time of day and geolocation of observations (Sofieva et al., 2014), absolute calibration (Frith et al., 2014),

and stability estimates and instrument uncertainty (DeLand et al., 2012). All of these factors, if not well accounted for, can introduce additional trends, uncertainties and errors, which may leak into statistical analyses of decadal trends (Tummon et al., 2015; Harris et al., 2015) and estimates of the magnitude of the response to drivers such as the Sun (Maycock et al., 2016). This can lead to conflicting results from different datasets (WMO, 2014).

Observational records of atmospheric ozone began with ground-based observations in 1921 (Staehelin et al., 1998) and were joined by satellites in the 1960s (Krueger et al., 1980). These records are an invaluable tool to understand not only the long-term trends in ozone, but also how the middle atmosphere operates. Ground-based observations have the advantage of being longer records and can be re-calibrated on a continuous basis, but they are point-source observations and thus cannot





account for large differences in ozone concentration and variability with latitude and longitude. The introduction of satellite observations have allowed for near-global, continuous observations over many decades, but has the disadvantages of typically only operating for a limited number of years and being subject to space-based degradation.

Creating an accurate record of stratospheric ozone profiles is a non-trivial task and much work has
been done at every stage, from design, construction and during flight, to post-processing and combining datasets into composites (Kyrölä et al., 2013; McPeters et al., 2013; Sofieva et al., 2013; Sioris et al., 2014; Froidevaux et al., 2015; Davis et al., 2016). Recently, several composites were published by multiple groups in connection with the SI2N initiative (SPARC (Stratosphere-troposphere Processes And their Role in Climate)/IO3C (International Ozone Commission)/IGACO-O3 (Integrated
Global Atmospheric Chemistry Observations – Ozone)/NDACC (Network for the Detection of Atmospheric Composition Change)) (Tummon et al., 2015). Nevertheless, even when problems are flagged, and uncertainties are minimized, the fact that different composites lead to trend estimates that differ by more than their uncertainties (e.g. Fig. 6 of Harris et al., 2015 and Fig. 8 of Tummon et al., 2015) means that at least one, if not all, are insufficiently stable during some periods to pro-
vide a robust estimate of changes in ozone throughout the stratosphere. Tummon et al. (2015) further notes that the choice of instruments to merge has more impact on trends than the merging technique used, that the construction approach needs careful consideration of the method used to avoid contaminating trends with artefacts, and that so far it has not been possible to remove biases from any individual, vertically-resolved, dataset.

Despite these issues, it is possible to account for many of these problems. There is common information within all the composites, e.g. the annual variability is similar in most composites (Tummon et al., 2015), and the differences between composite datasets due to the issues listed above should, in principle, point to where potential artefacts such as steps and drifts are located in time and by latitude and altitude. This can be especially effective in the case of an unexpected or erroneous change
occurring in one dataset, which is absent in all the others. Once the instrument or composite at fault is identified, there is the possibility of flagging, removing or rectifying an error, and confidence in applying a correction increases if the deviation or fault can be linked to a known issue. Thus, together with this prior knowledge and an unbiased uncertainty estimate, one can evaluate the likelihood of an observation being correct or, indeed, estimate the most likely value.

Our goal here is to provide a technique whereby the most likely ozone variability throughout the stratosphere can be identified by using the information embedded within multiple datasets simultaneously. The natural approach with which to tackle such a problem is using Bayesian inference (Cox, 1946; Lee et al., 2005; Arnold et al., 2007). If the solution were fully analytic, i.e. linear with known Gaussian uncertainties, then it would be straightforward to apply approaches such as
a Kalman-filter, or its more advanced varieties (see Arulampalam et al. (2002)). In this case, however, where composite datasets present non-linear and step-function changes, such an approach is





not sufficient. Thus, a non-linear, non-Gaussian approach is needed and a non-analytical solution will be necessary. A particle filter is one such approach (Pitt and Shephard, 1999); the method is named from the use of multiple point estimates of, in this case, ozone, which then propagate along

95 the timeseries and together form estimates of the likely (posterior) distribution at each time step. Specifically, we apply *sequential importance resampling* particle filtering (Pitt and Shephard, 1999; Arulampalam et al., 2002) following our own estimate of uncertainties in the composites along with prior information about seasonal variability and known problems within the underlying instrument data and the composites. This paper presents a test case for the application of particle filtering using

100 four ozone composites. It can be expanded to consider more observational records and, indeed, we recommend that it would be a logical next step to apply the technique to as many of the most up-to-date ozone datasets as possible. By its very nature as a Bayesian approach, further prior information about either known problems in the underlying datasets, or expected ozone variability, can also be included without effort.

105 This paper has three main parts. In the first (section 2), we introduce the composite datasets we use by explicitly presenting the problems we will later attempt to fix. Ozone composites have been updated since important inter-comparison papers by Tummon et al. (2015) and Harris et al. (2015), so our results cannot be directly compared with theirs; we briefly present some of these differences (section 2.2). The ozone composites, described in section 2, form a good starting point from which

110 to combine information and account for differences, since the effort put into producing them already considers and accounts for many instrument and observational issues. However, some remaining problems are clear in the composites. In the second part, we present the particle filter method to self-correct the ozone composites (section 3), construct uncertainty estimates (section 3.2), develop transition-priors to estimate how ozone is expected to vary on monthly timescales (section 3.3) and

115 discuss how we include additional prior information that we have available (section 3.2). Particle filter timeseries are presented and compared with the composites in sections 3.5 and 3.6. In the final part (section 4), we primarily use dynamical linear modelling (DLM) to evaluate long-term trends (section 4.2), although we compare our results with multiple linear regression (MLR) analysis, and present our results for ozone changes over the 1985-2012 period in section 4.3. We conclude in

120 section 5.

## 2 Ozone composites: background, updates and problems

### 2.1 Composites considered here

The SI2N project promoted seven ozone composites of satellite observations, summarised in Tummon et al. (2015), along with detailed comparisons that were expanded upon by Harris et al. (2015).

125 Three of the datasets, named SAGE-GOMOS1 (Kyrölä et al., 2013), SAGE-GOMOS2 (Penckwitt et al., 2015) and SAGE-OSIRIS (Adams et al., 2014) in Tummon et al. (2015), have more data





missing than the others (∼57% for 1985–2012 for 20°S—20°N), so we do not consider them in our analysis. Two of the remaining composites have the SAGE-II instrument (Stratospheric Aerosol and Gas Experiment II) (Damadeo et al., 2013) as a backbone: GOZCARDS (Global OZone Chemistry

And Related Datasets for the Stratosphere; Froidevaux et al. (2015)) and SWOOSH (Stratospheric Water and Ozone Satellite Homogenized; Davis et al. (2016)); we will refer to this pair of composites as 'SAGE-based'. The other two 'SBUV-based' composites we consider use the suite of SBUV-type (Solar-Backscatter Ultraviolet) instruments: SBUV-MOD (SBUV Version 8.6 Merged Ozone Data Set; Frith et al. (2014)) and SBUV-MER (SBUV Merged Cohesive; Wild and Long (2017)). By

using only two pairs of composites containing approximately equal weighting, we partly avoid the issue of biasing results to SAGE-based composites, a concern raised in the analysis of Harris et al. (2015) (though see appendix A3.4).

We consider zonal-, monthly-mean ozone over the 28-year period, January 1985 – December 2012, limited to the period covered by all datasets. While the correction method we present later

(section 3) could, in principle, be used to deal with data gaps at higher latitudes, we limit our latitude range to twelve 10° bands over 60°S—60°N. We limit the pressure range to 11 levels from 46–1 hPa (∼21–48 km) to avoid issues of large diurnal variations at higher altitudes, and because the vertically resolved SBUV data are not available at lower altitudes (i.e. at higher pressures). In order to treat each composite fairly, we interpolate all four onto the GOZCARDS pressure–latitude grid

since this grid has the lowest resolution of the four (though the instruments themselves have a higher resolution); a visualization of the original grids are shown in appendix Fig. A1. All considered composites have data available for more than 80% of all months at most latitudes. Finally, for this work, we are interested in relative variability and trends, so we shift absolute values to agree with the mean of SWOOSH from August 2005 to December 2012 when the Aura/MLS instrument is used;

during this period all the composites show remarkably good agreement on annual and multi-year timescales, and regression coefficients using multiple linear regression (see section 4.1) are similar at all pressure levels and latitudes (not shown). This is important since a common reference period we trust improves the particle filter's ability to estimate relative changes and reduces uncertainties.

The ozone instrument data and composites are already extensively detailed and discussed in sev-

eral recent papers as listed above, e.g. Tummon et al. (2015); Harris et al. (2015); we recommend that interested readers consult these, which also include an exhaustive list of references to individual instruments. We will discuss relevant points of interest regarding each composite in the discussion that follows below.

## 2.2 Why trend estimates differ between composites using the same data

To determine why decadal trends from the various composites are different, it is fundamental to understand how they have been constructed with satellite instrument data from multiple sources. We present a visual reference guide for the four composites in Fig. 1. Here, we show the time-





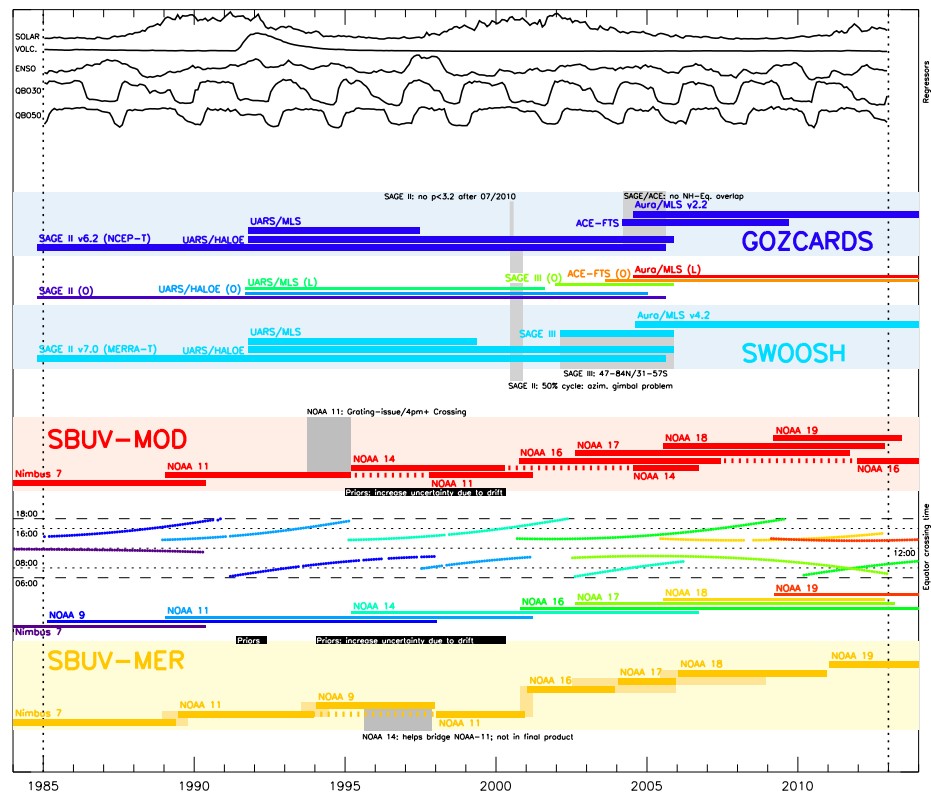

Figure 1: A guide to the regression indices used in the trend analysis (upper third) and instrument data used to construct SAGE-based (middle third: GOZCARDS, dark blue; SWOOSH, light blue) and SBUV-based (lower third: SBUV-MOD, red; SBUV-MER, yellow) composites. Shading at SBUV-MER instrument changes indicate periods used to determine differences in annual variability by applying bias corrections between instruments. The full periods of instrument operation for datasets in these pairs are shown with multiple colours between the composites. Where SBUV data are not used for an interval, dashed lines replace solid. Between the SBUV-composites, the local-time of equator crossing is shown. Where relevant, version numbers are given with instrument names; 'O' and 'L' indicate the satellite was a limb-viewer or occultation-based instrument; SBUV instruments are all nadir-viewing. Grey-shading with black text highlights periods discussed in the article. Periods specifically flagged to increase the SBUV uncertainty estimates in the particle filter are labelled black with white text.

line of instruments used to construct the SAGE-based data in the middle, and SBUV below. The colour-coding for the four datasets (GOZCARDS dark blue, SWOOSH light blue, SBUV-MOD red,



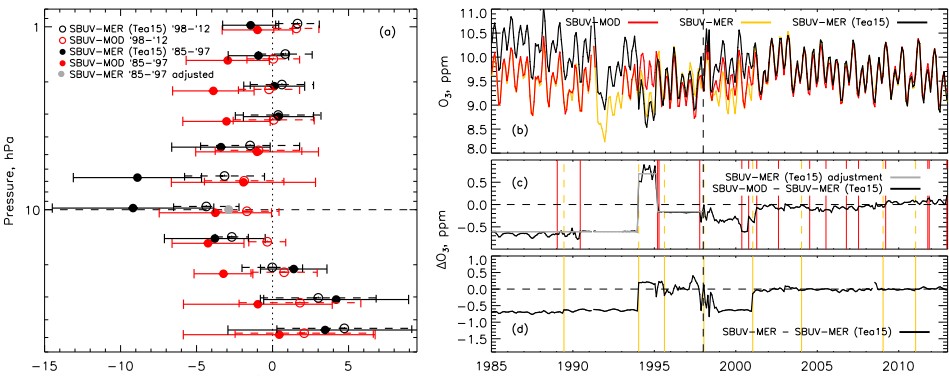

Figure 2: (**a**) The equatorial (20°S–20°N) decadal ozone MLR trend profiles for SBUV-MER version used by Tummon et al. (2015) ('Tea15'; black) and SBUV-MOD (red). Dots and solid error bars represent the 1985–1997 trends, and open circles and dashed error bars the 1998–2012 period. A single grey dot is plotted at 10 hPa, which follows an adjustment to SBUV-MER as shown in (c) as a grey line. (**b**) The ozone composite timeseries for SBUV-MER–Tea15 (black), SBUV-MER (yellow) and SBUV-MOD (red) at 10 hPa, all shifted to the July 2005 — 2012 mean of SWOOSH. (**c**) The difference between the SBUV-MOD and -MER-Tea15 timeseries in (b); the grey line prior to 1998 is a correction applied to SBUV-MER–Tea15 to produce the grey dot in (a). (**d**) The difference between SBUV-MER–Tea15 and SBUV-MER. The vertical dashed line in (b–d) indicates 1 January 1998, which delimits the two periods considered in the MLR results in (a). Error bars are $2\sigma$.

SBUV-MER yellow) will be used throughout the paper. The operating periods of all the instrument datasets used for either SWOOSH or GOZCARDS are presented as a spectrum of colours between them; the same is done for the SBUV-composites, where we additionally show information related to the time of day at which equator-crossings occur, which will be important later. Instrument names are given near the start of their operation period. Various comments and grey-shadings litter the plot;

these mark points to be aware of and some of these are discussed later.

### 2.2.1 SBUV-based composites

The main difference in how the SBUV-composites are constructed is that SBUV-MER considers only one dataset at a time (though it does use overlapping periods to account for absolute calibration biases and differences in seasonal and diurnal variation - shown with shading in Fig. 1), while SBUV-

MOD essentially averages overlapping data to combine them, relying on the instrument to instrument calibration done at the wavelength level within the version 8.6 algorithm for absolute calibration (i.e. no additional offsets are applied before averaging).

    The SBUV-based composites use only instruments with the same design and are the longest single-instrument-type composites available. Both use the same NOAA and Nimbus space-based

platforms, though not always at the same time, except that SBUV-MER uses NOAA-9 observations





between 1994 and 1997 to increase global coverage and bridge the gap in NOAA-11 (Fig. 1); this describes the updated version of SBUV-MER, which differs from the previous version considered by Tummon et al. (2015) (see below). The SBUV instruments infer profile ozone in units of parts per million (ppm) volume mixing ratio from measurements of back-scattered UV radiation at wave-
lengths shorter than 300 nm in a downward, nadir viewing system, which is fundamentally different from the limb/occultation instruments used in the SAGE-based composites; the SBUV instruments are optimised to low stray-light and high signal-to-noise radiance measurements, with an estimated accuracy of 1–2 DU at solar zenith angles up to 70 degrees (McPeters et al., 2013). Despite being constructed with essentially the same instrument data, the two datasets show differences in estimated
decadal trends (Harris et al., 2015; Tummon et al., 2015).

In Fig. 2a, we recreate the SBUV-MOD and SBUV-MER 1985–1997 (dots and solid lines) and 1998–2012 (circles and dashed lines) linear decadal ozone trend estimates from MLR (section 4.1) for the equatorial regions 20°S–20°N as in Figs. 5 and 6 of Harris et al. (2015) and Fig. 8 of Tummon et al. (2015). SBUV-MER has seen revisions since it was used in Harris et al. (2015) and Tummon
et al. (2015), so we use the version in those publications to make clear why previous analyses of the SBUV-composites differ (labelled 'Tea15'); after this section we only consider the latest update. The two composites show good agreement over the 1998–2012 period in both mean value and profile shape. The earlier period shows different vertical structure; at 10 hPa the mean values disagree by more than 5% per decade (the 10 hPa level is indicated by the horizontal dashed line). The reason
for this becomes obvious when we plot the absolute, and differences of, the timeseries at 10 hPa in Fig. 2b and c, respectively. Prior to 2002, the difference between SBUV-MER (Tea15) and SBUV-MOD can be almost as large as the annual variability. Fig. 2c reveals that these are caused by steps, of which the two largest occur in January 1994 and February–April 1995. We plot coloured vertical lines when instruments in either composite change (yellow for SBUV-MER, red for SBUV-MOD),
which immediately reveals that these jumps are related to offsets in instrument data: the first occurred in SBUV-MER, the second in SBUV-MOD. To prove it is these steps that cause the difference in the pre-1998 trend estimated at 10 hPa in Fig. 2a, we simply subtract the grey curve indicated in Fig. 2b from SBUV-MER (Tea15) and the mean MLR estimate for the trend is indicated as a grey-dot in Fig. 2a, now very close to SBUV-MOD. We note that this subtraction is not intended to indicate that
SBUV-MOD is correct, but is a simple test to understand why the trends differ.

Fig. 2d shows the difference between SBUV-MER (Tea15) and the updated version, which shows many of the offsets relative to SBUV-MOD in Fig. 2c have been removed. However, artefacts still remain in the newer version with respect to SBUV-MOD, and we find that they are not confined just to the altitude and latitude-range shown in these plots. Ultimately, the remaining differences lead
to the divergent trend estimates. We return to this in section 3.6; further discussion on the SBUV-composites is provided in Appendix A1.





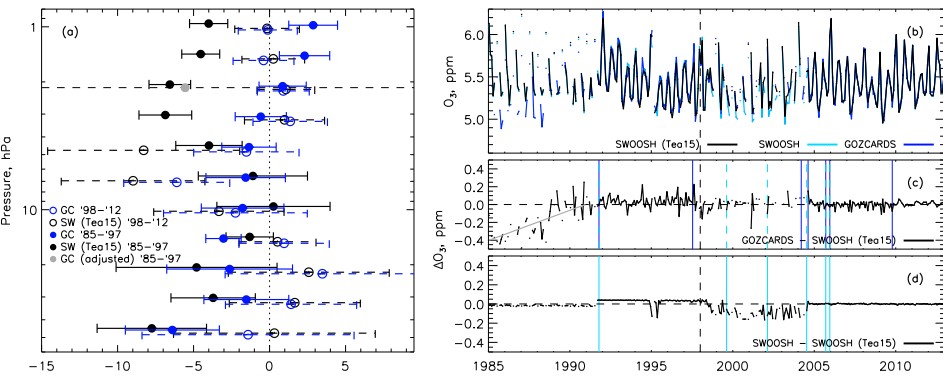

Figure 3: (**a**) The equatorial (20°S–20°N) decadal ozone MLR trend profiles for SWOOSH from Tummon et al. (2015) ('Tea15'; black) and GOZCARDS (blue). Dots and solid error bars represent the 1985–1997 trends, and open circles and dashed error bars the 1998–2012 period. A single grey dot is plotted at 2.2 hPa, which follows an adjustment to SWOOSH (Tea15) as shown in (c) as a grey line. (**b**) The ozone composite timeseries for SWOOSH–Tea15 (black), SWOOSH (light-blue) and GOZCARDS (blue) at 2.2 hPa, all shifted to the July 2005–2012 mean of SWOOSH v2.6. (**c**) The difference between the GOZCARDS and SWOOSH–Tea15 timeseries in (b); the grey line prior to 1991 is an adjustment applied to GOZCARDS to produce the grey dot in (a). (**d**) The difference of SWOOSH-Tea15 and SWOOSH v2.6. The vertical dashed line in (b–d) indicates 1 January 1998, which delimits the two periods considered in the MLR results in (a). Error bars are $2\sigma$.

### 2.2.2 SAGE-based composites

While constructed by two separate teams, GOZCARDS (Froidevaux et al., 2015) and SWOOSH (Davis et al., 2016) are similar for two main reasons: (i) the longest single instrument record used is SAGE-II (1984–2005) and this acts as the absolute reference level in both datasets; (ii) they are constructed from limb-viewers and occultation satellites (identified as 'L' and 'O' in Fig. 1), meaning they differ in operation from the SBUV-nadir viewers. Occultation satellites measure ozone number density on altitude levels looking at the disk of the rising or setting Sun though the atmosphere; this makes their vertical profile resolution higher, but at the expense of only observing 15 profiles per day. Limb sounders observe thermal emission in the infra-red or microwave as volume mixing ratio on pressure levels and can observe thousands of profiles each day. The composites differ in several ways, most relevant of which are: (i) they use different data screening and pre-processing; (ii) data from the same satellites are used for different periods and/or spatial regions; (iii) SWOOSH contains SAGE-III data and not ACE-FTS observations, and GOZCARDS vice-versa (see Fig. 1); (iv) GOZ-CARDS uses SAGE-II version 6.2, while SWOOSH uses version 7.0 – this innocuous difference has consequences for the trends (and solar signal analysis; not shown) that we will elaborate on in the following.





Because SAGE-II-based instruments observe ozone number density, knowledge of local temperature is needed to convert to volume mixing ratio. GOZCARDS uses SAGE-II v6.2, and SWOOSH SAGE-II v7.0; the former uses NCEP reanalysis temperatures while the latter uses the MERRA reanalysis (see Damadeo et al. (2013) and references within). It has been noted by McLinden et al. (2009), and confirmed by Maycock et al. (2016), that the NCEP temperature data contain spurious trends. The fact that the trend is not visible in SBUV data (section 3.6) further supports this. The impact of the different versions of SAGE-II within the SAGE-based composites is shown in Fig. 3. We note that, as for SBUV-MER, the current SWOOSH release has changed with respect to the aforementioned publications. Therefore, we again initially show results from the earlier version (2.1) in black (again designated 'Tea15'); following this discussion we will not refer to this version again. Fig. 3a shows the equatorial (20°S–20°N) decadal ozone trends similar to Fig. 2 extracted from GOZCARDS and SWOOSH (Tea15) using MLR for two periods: 1985–1997 (dots and solid lines) and 1998–2012 (circles and dashed lines). We see that for 1998–2012, except at 4.6 and 6.8 hPa, the two mean profiles agree well. However, for 1985–1997 above 5 hPa the ozone profiles show very large differences. To clarify why, in Fig. 3b we plot their 2.2 hPa timeseries, and their difference in Fig. 3c; the vertical dashed line indicates where the two periods considered in Fig. 3a are delimited. After 1991, both composites show similar long-term variability, though there are clearly sub-periods containing different scatter characteristics, and which change between instrument periods (vertical coloured lines), thus indicating a relationship to either different pre-processing or instrument usage. Between 1985 and 1991, GOZCARDS is lower than SWOOSH, and there appears to be an approximately linear increase over this period. Similar to the approach taken for SBUV-MER in Fig. 2, correcting the 1985–1991 period with a simple linear trend-line (grey in Fig. 3c) leads to very good agreement with SWOOSH (Tea15) in Fig. 3a (grey dot), showing the difference between the two SAGE-composites at 2.2 hPa is mainly caused by the pre-1991 trend in GOZCARDS; this is a result of the conversion of SAGE-II version 6.2 data (used in GOZCARDS) from densities to mixing ratios using NCEP temperatures, while the version 7 SAGE-II dataset (used in SWOOSH) uses MERRA and thereby corrects this issue.

Finally, we show in Fig. 3d the difference between SWOOSH (Tea15) and the latest version (2.6), which sees only minor step changes and short-term variance that appears to line up with instrument changes, except for between 1998 and 2004. Again, it is not clear from this difference plot alone if these changes lead to a better estimate of ozone variability and trends, or not. Further discussion on the SAGE-composites is provided in Appendix A2.

## 3 The particle filter

Additional problems in the composites can be revealed and understood by comparing SBUV- and SAGE- pairs. We will do this in section 3.6 alongside a newly constructed particle filter (PF) ozone





composite. In the following, we set out our methodology of how we can encode information as priors into the particle filter, with which we will then integrate the information to form the combined

composite.

### 3.1 Introduction to particle filtering

In order to combine the information from the various composites and correctly account for uncertainties, artefacts and drifts, we take a Bayesian approach and infer the posterior distribution $P(\mathbf{y}_t|\{\mathbf{d}_t\})$ of the true time-series, $\mathbf{y}_t$, given the full set of observations, $\{\mathbf{d}_t\}$. This also involves conditioning

on our knowledge about uncertainties and potential artefacts and drifts, and any prior assumptions about the month-to-month variability, all of which are combined into a background model, denoted $M$. Using Bayes's theorem, the posterior is given by

$$P(\mathbf{y}_t|\{\mathbf{d}_t\}, M) = \frac{P(\{\mathbf{d}_t\}|\mathbf{y}_t)P(\mathbf{y}_t, M)}{P(\{\mathbf{d}_t\}, M)}, \tag{1}$$

where the likelihood $P(\{\mathbf{d}_t\}|\mathbf{y}_t)$ summarizes our probabilistic model for the data given the associ-

ated measurement uncertainties (including our knowledge/assumptions regarding the possibility of instrumental artefacts systematically biasing the observations at certain times), the prior $P(\mathbf{y}_t|M)$ encodes our prior information and assumptions about the month-to-month variability of the underlying true time-series, and the marginal likelihood $P(\{\mathbf{d}_t\}|M)$ just plays the role of a normalizing constant here. Particle filtering, a form of sequential Monte Carlo sampling, provides a straightforward way

of drawing samples from the posterior conditional on all the data up to a certain point, from which we can reconstruct the full posterior density (given enough samples) or compute summary statistics (mean, variance, etc).

Particle filters (reviewed by van Leeuwen 2009 in a geophysical context) are useful for handling non-linear and non-Gaussian processes and have significant advantages over standard time-series

analysis techniques (Pitt and Shephard (1999); Arulampalam et al. (2002)). The essential idea is to propagate a set of 'particles' through the time-series in such a way that in the end they represent probabilistic realizations (i.e., samples) from the posterior distribution of the true time-series given the data. For example, each particle from the ozone (posterior) estimate of a given month is propagated forward using a transition prior encoding the anticipated change in ozone between that month

and the next. These prior samples are then re-weighted by an amount proportional to the likelihood of available observations; the new particles are then propagated to the next month again in a similar fashion. In the end the propagated particles constitute samples from the posterior of the true time-series given our prior assumptions and the observations up to whatever month is being considered. Under the assumptions of a Gaussian likelihood and prior, particle filtering reduces to Kalman filter-

ing. In the current work, however, a Gaussian prior and likelihood is not realistic (see section 3.2) so we require the more general particle filtering method.





In order to apply particle filtering to Equation (1) we require a probabilistic model for the data incorporating our knowledge and assumptions about the uncertainties, which in turn rests on a clear statement of our prior assumptions. We describe these below before giving a detailed description of
the particle filtering sampling algorithm in section 3.4.

**3.2    Likelihood and uncertainty estimation**

In order to treat each of the composites fairly, at least initially, we require uncertainties that reflect the possible deviation of the composite data from the true state of ozone at the time it was measured. We cannot use the uncertainties as published by the composite teams as they are not necessarily
derived in the same way or contain exactly the same information. The given uncertainty may or may not include: (i) uncertainties propagated at each step of the data and composite processing, e.g. in regression analysis used to combine individual instruments; (ii) uncertainties in the absolute offsets; (iii) the total number of observations in each dataset; or (iv) precision and calibration errors. A natural choice might be to use the number of observations used to form the monthly ozone value
from each instrument, but this is contentious because the number of data used does not mean that, e.g. a slow instrument drift is removed (e.g. SBUV instruments during the 1995–2000 period); if one simply considers using the number of data points to weight the monthly mean in each composite the mostly likely value would follow SBUV data almost exclusively until 2005 (see Fig. A2 in the appendix), and drifts would remain in the final product (see section 3.6).
Instead, we form our own uncertainty estimates using singular value decomposition. In principle, the timeseries at each latitude-altitude location in the four composites should be the same, and any deviations from the true value should be a result of one or more of the potential reasons listed in section 3.6. By this assertion, the composites each contain the real time series and an additional set of artefacts. The problem is that we do not know for sure in which dataset a problem might be,
especially if the true trend is only apparent in (or missing from) one composite, or one composite pair (i.e. SAGE- or SBUV-based). Thus, SVD allows us to separate the common signal from those that form the differences between the composites and the real ozone, leading to an attribution of higher uncertainty for single datasets that exhibit variance not present in the other three, and allows us to assign higher uncertainties in all the composites when one pair (e.g. SAGE-pair) acts differently to
the other pair (e.g. SBUV-pair).
Figure 4 helps explain how SVDs are used to attribute uncertainties. The left set of panels shows the SVD applied to ozone at 10 hPa 0–10°N: the SVD modes (black lines; first four panels) each have a different weight (percentage value in the lower right of each plot); the first mode contains most of the variance (84%) with the remainder split between the other three (13, 1 and 2%). The first
mode is common to all four datasets, and its relative weight within each dataset is represented by the colored dots to the right of each mode ranging from -1 to +1; the weight of mode-1 is similar in all four datasets. The second mode is split roughly equally between the two pairs of composites as



indicated by the dots on the right, suggesting it is the difference between the pairs, and for which the rescaled difference of SBUV-MER and GOZCARDS confirms, plotted in grey and with an almost

identical variance to the SVD mode. The SBUV-composites have almost zero weight in the third mode, indicating that the mode represents artefacts only within the SAGE-composites, again confirmed by the difference between SWOOSH and GOZCARDS (grey). With almost zero weight for the SAGE-pair in the fourth mode, the rescaled difference between SBUV-pairs confirms the mode represents artefacts in SBUV.

To form an ozone uncertainty from these, we ignore the first mode, as it provides no extra information due to its presence in all the datasets. Due to the separation and attribution of artefacts in the other three modes, we form an uncertainty estimate for each composite by weighting each mode by the respective weight in the composite (represented by the dots) and then taking the root of the sum of the squares of each weighted mode. This forms the uncertainty estimate for each of

the composites in the bottom panel. The SVD can only be formed when there is data available in each composite, which leads to gaps, represented by the grey shading in the bottom panel. Because composite sub-periods have different uncertainty-characteristics, we fill gaps using the median of the period between instrument changes in the composites (vertical lines in the four modes; colours relate to each composite).

The example at 10 hPa was ideal since modes were easy to associate with artefacts within and between the composite-pairs. Another example of the usefulness of applying the SVD to estimate the uncertainty is shown for 2.2 hPa and 0–10°N in the right-hand panels of Fig. 4b. The first mode is ubiquitous to the composites, and the fourth mode shows a clear attribution to the SBUV-composites (the rescaled difference is shown in grey). However, it is not possible to attribute modes two and three

as confidently, though the artefacts are more likely from GOZCARDS and SWOOSH, respectively. Since complete separation of this mode from the other composites is not possible (e.g. that SWOOSH is definitely the reason for the third mode), some uncertainty is given to the other composites. This is an intuitive approach to assigning uncertainty to each of the composites.

The error estimates satisfyingly display higher uncertainty to individual composites during periods

already known to have anomalous behaviour (section 3.6). For example, in the lower panel of Fig. 4 at 2.2 hPa (right), GOZCARDS is assigned a particularly high uncertainty during the first five years, as expected (section 2.2.2). At 10 hPa (left), the SBUV composites generally have a higher assigned uncertainty, especially around mid-1995, and until 2000, when we know there are instrument drifts in the SBUV-composites (section 3.6). Therefore, the SVDs allow us to independently and fairly

assign an uncertainty to each of the composites.

As the SVD is not always able to assign a known artefact explicitly to a specific composite, it is necessary for us to provide additional information regarding the composite uncertainties, whereby in three cases we increase the estimated uncertainty by a factor of two. These are: (i) when an instrument changes in a composite, which is appropriate since there are many examples of jumps in





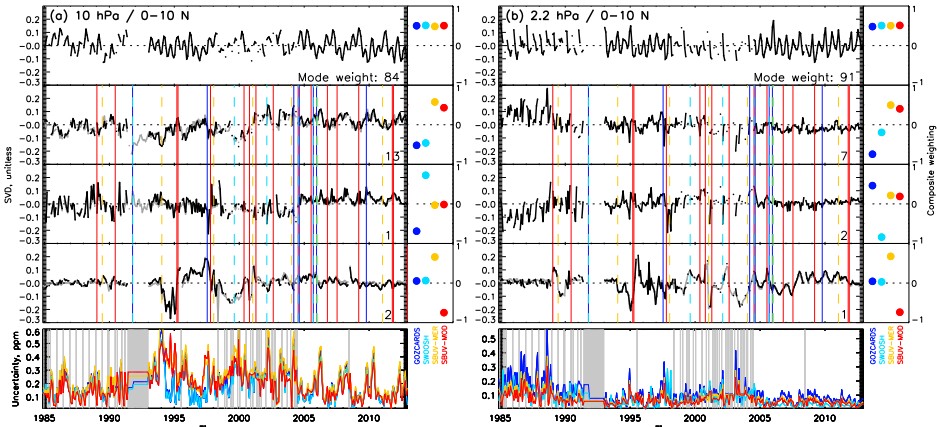

Figure 4: Visualisation of the process used to estimate the uncertainty on each ozone composite for two examples at (**a**) 10 hPa and (**b**) 2.2 hPa at 0–10°N. The left column of the first four rows show the determined singular value decomposition (SVD) unitless modes (black time series); the mode weighting (%) is given in the bottom-right; the right column is the mode weighting for each composite. All colours represent information related to GOZCARDS (blue), SWOOSH (light-blue), SBUV-MER (yellow) and SBUV-MOD (red). Vertical lines represent dates an instrument change in the composite occurred. The grey timeseries is the arbitrarily rescaled difference between SBUV-MER–GOZCARDS, SWOOSH–GOZCARDS and SBUV-MER–SBUV-MOD in (a) in rows 2–4, and SBUV-MER–SBUV-MOD in (b) in row 4. The bottom panel (row 5) in (a) and (b) represents the uncertainty derived from the root sum of the squares of the modes (rows) 2–4, weighted by the mode and composite weight, in units of ppm. Grey vertical lines represent dates when data in any composite is missing and filled with the median uncertainty for the sub-period in which they lie (i.e. between the vertical lines in rows 2–4).

a composite on or immediately after these dates (e.g. Fig. 2c in 1994 and 1995); (ii) during known and significant instrument drifts in SBUV – the SBUV drift from the SVD uncertainty estimate is typically assigned equally to both pairs of composites and so additional information is needed and tests show that it is only partially accounted for when this additional information is not included – specifically 1995–2000 for both SBUV-composites, and additionally 1994–1995 in SBUV-MER

(these periods are marked by black shading and white text in Fig. 1); (iii) following the eruption of Mt. Pinatubo in SBUV-MER only (see Fig. 1 and section 3.6).

With the uncertainties on each composite estimated, we construct a joint-likelihood function for the set of composites as $P(\{\mathbf{d}_t\}|\mathbf{y}_t) = \prod_t P(\{d_t\}|y_t)$, where we take the joint-likelihood for a given





time-step as:

$$P(\{d_t\}|y_t) = \prod_c \frac{1}{\sigma_t^c \sqrt{2\pi}} \left\{ \frac{\beta}{\gamma} \exp\left[ -\frac{(d_t^c - y_t)^2}{2\gamma^2 \sigma_t^{c\,2}} \right] + (1-\beta)\exp\left[ -\frac{(d_t^c - y_t)^2}{2\sigma_t^{c\,2}} \right] \right\}, \qquad (2)$$

where $d_t^c$ is the data at time $t$ for composite $c$, $y_t$ is the underlying time-series we are trying to infer and $\sigma_t^c$ are the uncertainties for composite $c$. The likelihoods for each individual data point also contain constants $\gamma$ and $\beta$ which characterize our level of suspicion that a given data point is an outlier or unreliable (e.g., systematically biased due to some instrumental artefact). This form allows bimodality in the product of individual likelihoods which is of central importance (Box and Tiao, 1968). In cases where the composites disagree due to an artefact in one or more dataset, simply multiplying Gaussian likelihoods together would result in a joint-likelihood that sits between the two (or more) peaks and does not represent likely values according to any of the composites. However, under the model prescribed by Equation 2, the joint likelihood is multimodal where subsequent application of the prior may elicit which of the peaks is representative of the truth and which observations were likely dominated by artefacts (or indeed if all composited might be systematically biased simultaneously but in different ways, in which case the resulting posterior for that point will have an inflated uncertainty as desired). The parameters $\gamma$ and $\beta$ must either be fixed by hand (where smaller values of $\beta$ encode less faith in individual observations) or kept as hyper-parameters; we fix $\gamma$ and $\beta$ to give multimodal behavior as desired.

### 3.3 Transition-prior

For the prior assumptions, we factorize the prior into a product of transition-priors for each month-to-month transition, i.e.,

$$P(\mathbf{y}_t) = P(y_0) \times \prod_{t=1}^{N-1} P(y_{t+1}|y_t). \qquad (3)$$

The usefulness of the transition-prior is to place particles in approximately the right region of the state-space for that month; it is not a strongly constraining property of the particle filter, but provides a way to estimate if distributions of ozone values from the composites in the month being evaluated are more likely or not, and in particular to provide a way of assessing anomalous behavior. The annual, or semi-annual, variability that makes up the seasonal cycle, is the largest mode of ozone variability. It is also a relatively consistent mode, so together with information from the observations, it can provide a way to help differentiate between artefacts and real anomalous behavior.

We form the transition prior from all four composites together. Two examples are given in Fig. 5 at 2.2 and 10 hPa at 0–10°, where the expected change between month n and n+1 for the whole year is shown, with e.g. n=1 being the transition between January and February. The monthly changes for all composites are shown with the box-and-whisker plots, which show the mean (white horizontal line), inter-quartile range (IQR, 25–75th percentiles; thick stem) and full range or 1.5 times the IQR (thin line), with any outliers given as dots. The grey Gaussian distributions are formed from



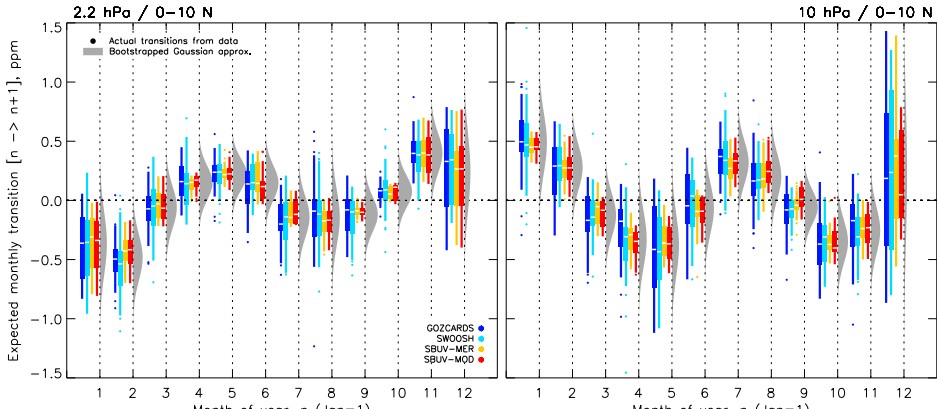

Figure 5: The expected monthly ozone changes (or 'transitions') between month n and the next month, n+1, i.e. index 1 represents a change between January and February. We show two examples at 0–10°N: (left) 2.2 and (right) 10 hPa. The box-and-whisker plots are for all observations when no change in the underlying instrument of the composites occurred and represent the inter-quartile range (IQR) covering the 25th to 75th percentiles (box) and 1.5 times the IQR or the maximum, whichever is smallest (whisker); outliers are plotted as dots. Plotted to the left of the vertical lines at each index are the changes between months for each composite (represented by the different colours); Gaussian distributions to the right of the vertical lines represent those formed from the mean and standard deviation of all the composite transitions from 1000 bootstraps. These Gaussians are used as transition-prior estimates and are calculated for all pressures and latitudes.

all the changes between two months treated independently and then performing 1000 bootstraps. We note that in the examples shown in Fig. 5, the SAGE-based composites typically have a larger range of month-to-month variance, which we suggest may be due to the higher resolution of the SAGE-composite instruments, but we cannot exclude the possibility that this is also related to the low-sampling and higher scatter of, e.g., the earlier observations from SAGE-II.

### 3.4 Particle filtering for drawing posterior samples

We elucidate the particle filter with a real example at 2.2 hPa (each step is also written out in Algorithm 1 of the appendix): the 'preparation' step (Algorithm 1) is assumed to be complete and involves prescribing the prior information and calculating the transition-prior in advance; all composites are initially shifted to the mean of SWOOSH over the 2005–2012 period where they all show similar behaviour and stability.

At the first month in the time-series, January 1985, there is no prior information available (step 1), so in Fig. 6a we generate N particles drawn from a uniform prior of width ±2 ppm, represented as a



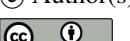

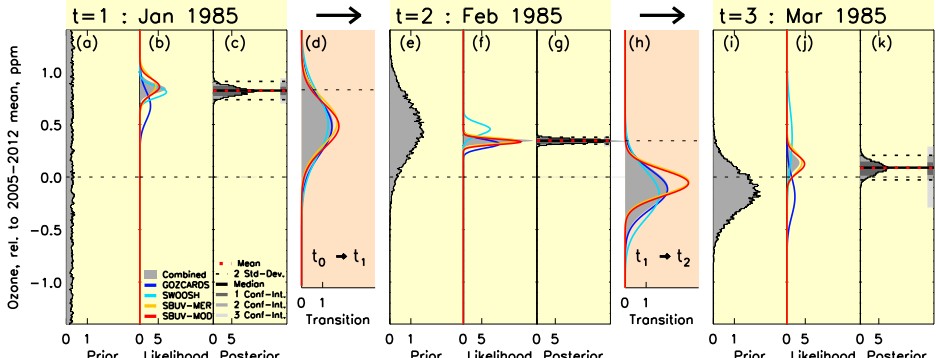

Figure 6: Graphical representation of the particle filtering approach, considering the first three time steps between January and March 1985 at 2.2 hPa and 0–10°. Here, the plotted range is relative to the 2005-2012 mean of SWOOSH, to which all datasets have been shifted. Note that expected 'transitions' between each month plotted in (d) and (h) are the expected ozone change between months, not the absolute ozone estimates given in all the other plots and the zero line in (d) and (h) is the dotted line centred on the median (peak) of the distribution in (c) and (g) respectively. The legends provide information about the different parts of the plots and a detailed explanation of the figure is given in section 3.4. All distributions are normalised to have an area equal to one, however note the scales are difference in each panel.

distribution (grey shading). The coloured lines in Fig. 6b show the individual composite likelihoods as Gaussian distributions. Any additional prior information about jumps or drifts is incorporated here by increasing individual composite uncertainties by a factor of two (step 2; see section 3.2). The information in the composites is then combined to form the joint likelihood from Equation (2).

Next, we need to consider carefully how to weight the particles in the prior. It is typical that, if we simply weight the particles by the posterior distribution, within a few time steps only a few particles have most of the relative weight in the distribution (van Leeuwen, 2009). To avoid the associated degeneracy we resample the particles in the prior by the likelihood so that each position in the posterior distribution is then represented by multiple particles of equal weight, instead of a

single particle of high-weight. The advantage of this is that each of the many particles near regions of high-density in the posterior have an opportunity to spread out in the next month, rather than all the weight being concentrated in one region represented by just a few particles, or one. This particular type of particle filtering is known as *sequential importance resampling* (SIR). The result is that particles in Fig. 6a far away from the peak in Fig. 6b will receive very little weight and will be

unlikely to survive into the posterior in Fig. 6c. The re-weighting of particles (Step 4) is performed so that the particles form representative samples of the posterior distribution.





The re-weighting is performed by assigning a weight to each particle in the prior with the probability in the likelihood. The weights of the particles are then ordered and each particle is resampled from the ordered distribution by randomly sampling from a uniform distribution and selecting the particle within the sorted distribution that the random value falls on. In this way, particles with higher weights are resampled more frequently and thus the posterior distribution represents the prior multiplied by the likelihood, i.e. the posterior in Fig. 6c. We use a large number of particles and re-sampling to prevent a degenerate collapse of the distribution to just a few particles, or one; we have not observed this occurring in our experience of the SIR particle filter in this setup. The posterior formed in Fig. 6c represents our full knowledge of ozone in January 1985 given the prior information and the observations available to us. If more information or more data becomes available, we can encode this in the prior or the likelihood, respectively, to improve the inference. From the posterior we can produce summary statistics, which we provide a few of here, and in the product release[1]. For example, the mean and median are shown in Fig. 6c as a dotted red and solid black line, respectively. Two-standard deviations are shown by the dotted black lines, but the posterior is typically not Gaussian, so the 68, 95 and 99% credible regions are shown by the thick dark-grey, grey and light-grey bars, respectively, to the right of the posterior.

In the next step (5), we transition the posterior to February 1985 using the transition-prior in Fig. 6d (which is the same as in Fig. 5 at index 1 in the left plot); for reference we indicate the zero line as a dotted line centred over the median of the posterior peak in Fig. 6c. To form the prior samples in Fig. 6e, we sequentially select a particle in the posterior from Fig. 6c, and add a random sample from the transition-prior in Fig. 6d. When all particles are transitioned, the prior in Fig. 6e is formed. The next step (6), and all subsequent steps until the completion of the timeseries, repeat steps 2–5. So, by example, in Fig. 6f, we form the likelihood using Equation 2, re-weight the particles in the prior in Fig. 6e with this likelihood and resample the sorted-weighted distribution to form the posterior for February 1985 in Fig. 6g. This forms the basis for the prior in March 1985 (Fig. 6i) using the transition-prior in Fig. 6h, and so on, until complete. This process can be repeated with multiple initialisations, but we have found that a single run of 10,000 particles leads to a robust estimate of the posteriors and is sufficient.

Repeating this procedure at every altitude and latitude leads to a new particle filter ('PF') composite.

## 3.5 Particle filter example results

We designed artificial tests to evaluate whether the particle filter was effective in retrieving the 'true' ozone timeseries given a set of four ozone composites that had jumps, drifts and noise, similar to those we encounter in the existing datasets. Overall, we found the particle filter to be successful

---

[1]The PF results will be made available online upon publication of the paper.





at estimating ozone, better than any individual composite that contains artefacts. These tests are presented in Appendix A3.3.

The full Particle Filter (PF) composite result for the 0–10°N 2.2 hPa timeseries is given in Fig. 7a, with all four composites, and the PF with uncertainties at two-standard deviations (dotted lines) and
68, 95 and 99% confidence intervals (dark, medium and light grey shading); the differences in Fig. 7a relative to the PF are shown in Fig. 7b. It is clear that the PF has successfully accounted for (i) the early drift prior to 1991 in GOZCARDS resulting from the use of NCEP reanalysis temperatures, and (ii) the high scatter in both the SAGE-composites prior to 1991 and mainly in SWOOSH prior to 2004 resulting from the low-sampling of the occultation instruments used. When disagreement
between composites increases, or the priors inflate the uncertainties, the PF uncertainty estimate naturally inflates to allow for the higher uncertainty during that period; on the other hand, the PF uncertainties reject most of GOZCARDS prior to 1989 at the 99% confidence level.

Another example, at the higher pressure of 10 hPa, is given in Figs. 7c and d. Here we see that the PF has accounted for (i) the SBUV-MER problem following the Mt. Pinatubo eruption, during which
SBUV-MOD measurements are not provided, (ii) rapid steps in the SBUV-composites between 1995 and 2001, and (iii) some of the drifts in the SBUV-composites during the same period. What is clear here, especially in the period after 2002, is that while the PF reproduces most of the variance, it cannot determine whether the higher amplitude variance of the QBO signal in the SAGE-composites is more likely to be correct than the SBUV-composites, though we know the reason is due to the
lower vertical resolution of the SBUV-type instruments and that the QBO represented by the SAGE-composites is more likely to be correct (see section 3.6). We do not currently have a solution for this particular issue, though the errors do inflate naturally to accommodate this uncertainty, and so within the uncertainties this issue is captured by the PF.

Finally, to show how the PF operates in a completely different regime to that near the equator,
in Fig. 7e and f we give an example at 6.8 hPa and 50–40°S. Here, ozone lacks a semi-annual component of variability. Except for between 1993 and 2001, all four composites show broadly similar variability. The SAGE-composites again appear to show spikes that aren't present in the SBUV-composites, and indeed on many occasions do not occur in both SAGE-composites. There-fore, many of these are rejected by the PF. We cannot discount that some of these artefacts are a result
of the better resolution in the SAGE-composites and may be real, for example unexplained artefacts after 2008, but these are generally found to remain at or within the 95% confidence level. Follow-ing the instrument change in SBUV-MER in 1994, and until 2001, we see anomalous behaviour in SBUV-MER that is rejected by the PF at the 95% level throughout this period; between 1995 and 1997, SBUV-MOD also displays behaviour quite different to the other composites, and this is also
generally rejected.



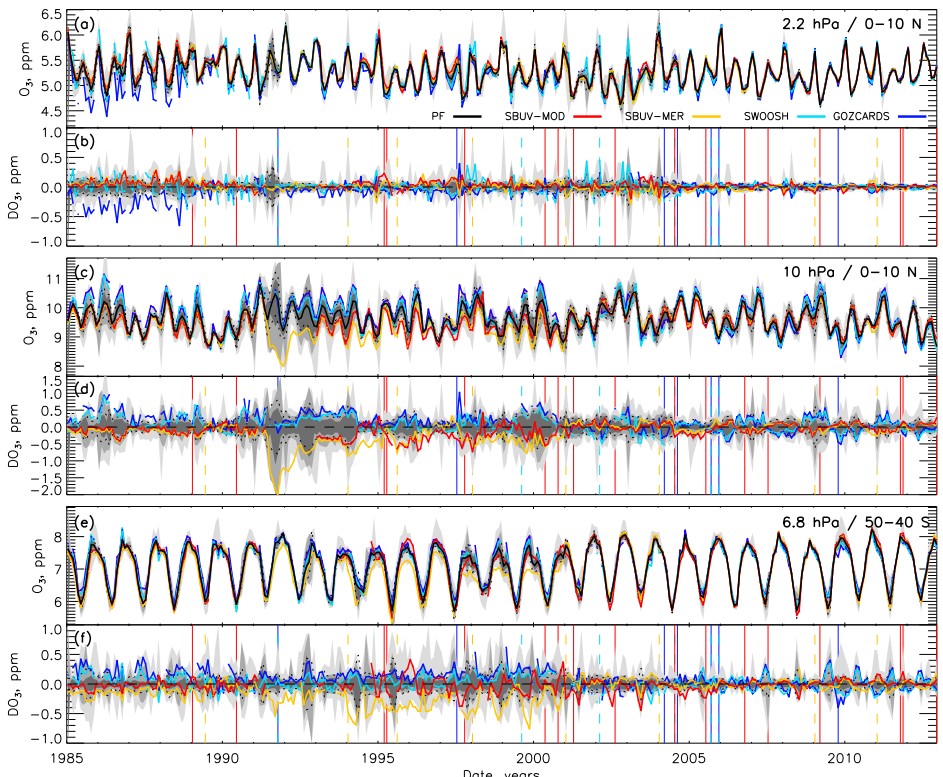

Figure 7: Ozone time series at three stratospheric locations from 1985–2012, all bias-shifted to the mean of SWOOSH after August 2005. (**a**) Absolute ozone at 2.2 hPa over 0–10°N from SWOOSH (light-blue), GOZCARDS (blue), SBUV-MER (yellow), and SBUV-MOD (red). The particle filter median estimation (black) is plotted with shading representing 68% (dark-grey), 95% (grey) and 99% (light grey) confidence intervals (CIs); these CIs are not Gaussian, so 2 times the standard deviation is also plotted with thin dotted lines. (**b**) As for (a), but now the difference relative to the particle filter median. (**c**) and (**d**) are as for (a) and (b) at 10 hPa and 0–10°N, and (**e**) and (**f**) are as for (a) and (b) at 6.8 hPa and 50–40°S. Vertical dashed and solid lines in (b), (d) and (f) identify changes in the instruments used in the composites.

## 3.6 Detailed examples of differences in all four composites being resolved by the particle filter

In sections 2.2.1 and 2.2.2, we showed examples of differences between composites based upon the same, or similar, instrument data. It is not always clear by looking at the pairs of composites, however, which is more likely to be correct: drifts and rapid changes occurring over a few months cannot be immediately attributed to a specific composite. However, as we will now demonstrate,


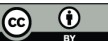


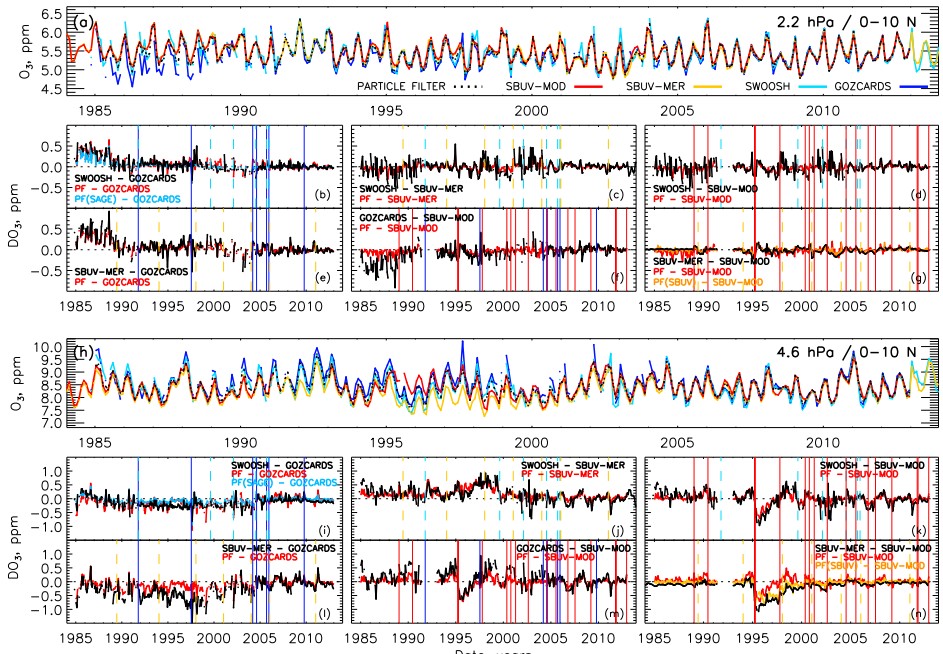

Figure 8: Ozone time series at two stratospheric locations from 1984–2014, all bias-shifted to the mean of SWOOSH after June 2005. (**a**) Absolute ozone at 2.2 hPa over 0–10°N from SWOOSH (purple), GOZCARDS (blue), SBUV-MER (yellow), and SBUV-MOD (red). (**b-g**) The difference between each pairing of the four composites (see legends). (**h–n**) As for (a–g) but at 4.6 hPa and 0–10°N. Solid and dashed vertical lines represent months with a change in the instrument used to construct the composite (colours are with respect to the composite colour in (a) and (h).

additional information from the literature, knowledge of when instruments are added or removed within the composites, and looking at the differences of all four composites at the same time, helps to build confidence in attributing the source and reason for the deviation, and then correcting it – these are encoded in the uncertainties of each composite as discussed in section 3.2. We also show

the effectiveness of the PF in accounting for most of these artefacts. The final PF ozone composite product that integrates information from all four composites is denoted 'PF'. But, it is also possible to only use information from either the SBUV-pair ('PF(SBUV)') or SAGE-pair ('PF(SAGE)') of composites, which elucidates how the prior information applied in the particle filter algorithm is able to perform if information is missing from the other pair of composites. In other words, a correction

of artefacts (e.g. drifts in SBUV-composites) that do not appear in differences of just one composite pair strengthens our claim that the PF is correctly accounting for artefacts in the composites. For clarity in the figures introduced here, we do not provide uncertainties on the PF.





In Fig. 8 we show two examples of the four ozone composites at 0–10°N, at 2.2 hPa (a–g) and
4.6 hPa (h–n). Below the absolute timeseries (a and h) are six plots (b–g and i–n), which are the
differences between each pairing of composites (black); the absolute PF composite ozone is shown
with a dotted line, and differences of the PF, PF(SAGE) and PF(SBUV) compared to the composites
are given in red, blue and orange, respectively. Once again, the early trend (e.g. (b) SWOOSH –
GOZCARDS) and the steps (e.g. (n) SBUV-MER – SBUV-MOD) are clearer in these restricted
latitude bands than in the broader equatorial band presented in Figs. 2c and 3c. However, considering
these different pressures and latitudes, and the SBUV–SAGE differences (c–f), additional anomalous
behaviour is revealed, which we list and discuss in the following.

1. The most significant problem in creating a unified calibration for all SBUV instruments is
the orbital drift (McPeters et al., 2013). Ideally, the local time at equator crossings should be
the same each orbit, and near-polar to attain near-global coverage. However, NOAA satel-
lites slowly drifted over time, changing from near 2pm (10am, NOAA-17) equator crossings
to late afternoon (early-morning, NOAA-17) equator crossings. NOAAs 9, 11, 14 and 16
drifted through the terminator and began making early morning measurements. The equator-
crossing time for each of the SBUV satellites is shown in Fig. 1 between the SBUV-MOD
and -MER composite information. Any instrument or calibration errors may be significantly
enhanced for observations taken as the orbit approaches the terminator, such that the orbit
drift can lead to an apparent time-dependent trend in ozone that could be misinterpreted as
real; McPeters et al. (2013), DeLand et al. (2012) and Bhartia et al. (2013) do not recommend
the use of near-terminator data for this reason. Accordingly, SBUV-MOD, with the excep-
tion of NOAA-11, does not include any observations taken outside the 8am–4pm equatorial
crossing time range (marked as dotted horizontal lines in Fig. 1) and similarly SBUV-MER
prioritizes measurements made while instruments are in their optimum orbits. The clearest ex-
ample of this drift related trend can be seen in Fig. 8k, m and n in all differences with respect
to SBUV-MOD between 1995 and 1998 (until 2000 with respect to SBUV-MER in Fig. 8n);
there is then a reversed trend until after 2000. The differences with the SAGE-composites in-
dicate that: a 1994–1995 drift is likely in SBUV-MER from the exclusive use of NOAA-9;
for 1995–1997 the drift is probably in both, but more prominent in SBUV-MER differences;
the 1997–2000/2001 drift is more likely in SBUV-MER with the exclusive use of NOAA–11
(SBUV-MOD merges NOAA-11 with NOAA–14). Other, smaller drifts between the SBUV
composites are visible in (n), e.g. in 2001 and 2002. While the PF(SBUV) and PF were able to
account for the large discontinuity present in Fig. 8n, the PF(SBUV) is unable to account for
the 1997–2000 drift in SBUV-MOD. We do inform the particle filter routine that the uncer-
tainties should be increased in the SBUV-composites during the drift period 1995–2000 (from
1994 in SBUV-MER), so uncertainties are equal for this period in the PF(SBUV). Neverthe-
less, with the inclusion of the SAGE-composites this drift can be accounted for (red line in



Fig. 8j–n), which further reinforces the need for information from all composites to resolve problems. Confirmation of drift problems during the periods mentioned (DeLand et al., 2012; McPeters et al., 2013; Kramarova et al., 2013b; Frith et al., 2014) justifies using it as prior information to down-weight these data for this time (see Appendix A1 for more information).

2. The apparent high scatter at 2.2 hPa in all differences involving SAGE-composites (i.e Figs. 8b–f) during the periods of 1985–1991 and 1997–2004 coincides with periods when only occultation instruments were active (SAGE-II, UARS/HALOE and ACE-FTS). Toohey et al. (2013) and Sofieva et al. (2014) convincingly demonstrated that insufficient and/or inhomogeneous sampling can result in inaccurate monthly estimates and even induce spurious spikes in ozone timeseries; coarse-sampling occultation-type instruments such as GOMOS/OSIRIS and ACE-FTS, can lead to differences of up to 20%. This can especially affect seasonal cycle representation, especially at high altitudes where ozone undergoes rapid variations with latitude and time of day. This is why spurious variability from occultation instruments is clearly evident in Fig. 8 during the aforementioned periods. Even though satellite measurements from limb-viewers have a lower vertical-resolution than occultation, these are still sufficient to reduce the monthly zonal-mean scatter in the SAGE-based composites when overlaps with occultation instruments occur (e.g. 1992–1997 in GOZCARDS). The PF–GOZCARDS difference in Fig. 8e agrees closely with the month-to-month artefacts that are highlighted in the SBUV-MER–GOZCARDS difference. This is not because of the information provided in the SBUV-composites, which don't display this behaviour, but because the deviation from the natural seasonal cycle is so high that the month-to-month seasonal variability is more informative. This is confirmed by the high agreement between PF–GOZCARDS with PF(SAGE)–GOZCARDS on these short-timescales in Fig. 8b, the latter of which contains no knowledge from the SBUV-composites.

3. The downward trend between the SAGE-composites prior to 1991 (Figs. 8b and i; see section 2.2.2) is largely absent in the SWOOSH composite compared to SBUV-composites (Figs. 8c and d), confirming it as a feature of GOZCARDS only. It is clear from Fig. 8e that the trend in GOZCARDS prior to 1991 is fully accounted for by the PF, and once again the agreement of PF–GOZCARDS with PF(SAGE)–GOZCARDS in Fig. 8b shows that the information in the SAGE-composites alone is sufficient to eliminate most, though not all, of this problem. No prior information about the trend being in GOZCARDS is provided to the particle filter – the ability for the PF to account for the trend is most likely because SWOOSH agrees with the prior information from the seasonal variability (in the transition prior) much better than GOZCARDS.

4. A small downward step in the SAGE-composite difference in Fig. 8b and i in 2004 occurs around the time both SAGE-composites have an instrument change. This feature is more evi-





dent in the differences between GOZCARDS and the SBUV-composites than for SWOOSH, at both altitudes. At the lower altitude of 4.6 hPa in Fig. 8i, it appears that PF(SAGE) could not account for the jump in GOZCARDS, and ends up slightly offset from the black difference line. The PF performs better with the additional information provided by the SBUV-composites, and fully accounts for this jump.

5. A prominent feature in Fig. 8j–m is the approximately 2–3 year oscillation. This is the result of lower vertical resolution in the SBUV observations, which leads to a damping of the quasi-biennial oscillation (QBO) signal in SBUV relative to the higher resolution instruments of the SAGE-based composites; at 3 hPa SBUV has a vertical resolution of approximately 6-7 km, while the SAGE-based instruments are usually better than 3.5 km - the vertical resolution only gets larger for SBUV with lower altitude, reaching a maximum of ~15 km below the tropopause (Bhartia et al., 2013). After 2003, the resolution effect is more clearly visible in Fig. 8h, since many of the other instrument-data/composite artefacts are absent. Kramarova et al. (2013a) showed that by applying the SBUV resolution kernel to higher vertical-resolution Aura/MLS data led to good agreement with SBUV data. Focusing on the period after 2005 in Figs. 8h–n, it is evident that the PF is unable to distinguish between the QBO represented in the SAGE- and SBUV-composites; this is because uncertainties are similar during this period and composite issues are generally absent. We discuss this further in Appendix A3.4.

6. Following the eruption of Mt. Pinatubo in June 1991, there is a large drop in SBUV-MER at 10 and 16 hPa due to interference in viewing from volcanic aerosols (not shown here, but see Fig. 7c and d), which is absent in the SAGE-composites; SBUV-MOD does not include data during this period. Ozone is usually depleted by sulphate aerosols following a volcanic eruption, but at lower altitudes. Due to the rapid departure of SBUV-MER from the SAGE-composites, the PF predicts that the SAGE-composites are more likely to be correct during this period. To be clear, the PF can adapt to rapid, unexpected changes in ozone: if all the datasets had shown a sudden, and similarly large change that was significantly different from the prior expectation for that month, it would tend towards a tighter cluster of observations as more likely than the broader prior estimate. The only clear problem we find for the PF following the eruption is at 15 hPa during the first 12 months following a change of instruments in the SAGE-composites; this leads to a higher uncertainty in all datasets when SBUV-MER is recovering to levels similar to the SAGE-composites and ozone levels prior to the eruption. We discuss potential remedies to this in appendix A3.4.

7. For completeness, steps in the SBUV-composites in Figs. 8k, m and n, discussed in section 2.2.1 occur in 1995, and in 2003, 2004 and 2007 in n; though these are not the only times that steps occur; prominence of steps depends on altitude and latitude. The PF accounts for these discontinuities, which is most clear for the large jump in the SBUV–MOD composite in





Figs. 8k, m and n; absence of a jump in Fig. 8i confirms the PF's success. For the PF(SBUV)–
SBUV-MOD case in Fig. 8n (orange), which relies exclusively on the SBUV-composites, the
large step in 1994/1995, and drift that follows, is mostly accounted for.

## 4   Analysis of ozone data

Now that we have established the validity of the particle filter approach, and constructed an ozone
composite from GOZCARDS, SWOOSH, SBUV-MOD and SBUV-MER, we turn to analysing
trends and modes of variability. This is often performed using multiple linear regression (MLR)
(WMO/UNEP, 1994; Soukharev and Hood, 2006; Chiodo et al., 2014; Kuchar et al., 2015; Harris
et al., 2015). However, the use of Dynamical Linear Modelling (DLM), first applied to ozone data by
Laine et al. (2014), appears to be more robust at estimating the background trend, especially if it is
non-linear. Laine et al. (2014) noted this when comparing their DLM results with the MLR results of
Kyrölä et al. (2013) where linear trends were sometimes found to be inverse to those estimated using
DLM. We performed tests upon the artificial timeseries used to evaluate the performance of both
methods with the particle filter (section 3.5 and appendix A3.3). We briefly introduce both methods
below. We compare their performance on the artificial timeseries and the particle filtered correction,
introduced in Appendix section A3.3, in Appendix section A4. We found that in every test-case the
DLM did equally well, or better, at estimating the true background 'trend' than the linear estimate
from MLR (see appendix Figs. A4 and A5), both for non-linear background trends and for timeseries
with large artefacts.

### 4.1   Multiple linear regression (MLR) analysis

We perform MLR analysis on deseasonlised timeseries (i.e. by subtracting monthly means) using
five regressors: the F30 radio flux (solar), which is superior to the F10.7cm radio flux for represent-
ing solar UV variability (Dudok de Wit et al., 2014); the stratospheric aerosol optical depth (SAOD
(Sato et al., 1993), for volcanic eruptions); the El Nino Southern Oscillation (ENSO); and two or-
thogonal modes of the dynamical quasi-biennial oscillation (QBO). These regressors are displayed
in the upper part of Fig. 1. When we analyse decadal trends between 1985–1997 and 1998–2012,
we use a linear trend to estimate the long-term trend. We use pre-whitening and a first-order autore-
gressive process (AR1) to account for auto-correlation in the residuals (Tiao et al., 1990). Statistical
significance of the regression coefficients was evaluated with a Student's t-test.

### 4.2   Dynamical Linear Modelling (DLM)

We perform a Dynamic Linear Modeling (DLM) analysis following very closely the model and for-
malism of Laine et al. (2014). We use the same five regression components as in the MLR. We allow
for two modes of seasonal variability in the fit (with 6- and 12-month periods), where additional





(Gaussian-process) variability of the sinusoidal seasonal modes is also allowed for (following Laine
et al. (2014)), and variance of the (Gaussian) seasonal model-variability $\sigma_{\mathrm{seas}}^2$ is kept as a free param-
eter in the fit. We include an AR1 process, where the variance $\sigma_{\mathrm{AR}}^2$ and correlation coefficient $\rho_{\mathrm{AR}}$
of the AR1 process are also kept as free parameters in the fitting process. In contrast to MLR, the
DLM approach allows for a fully non-linear 'trend', where the degree of non-linearity $\sigma_{\mathrm{trend}}$ is also

kept as a free parameter in the fit (see Laine et al. (2014) for details). In further contrast to MLR, the
Bayesian DLM approach jointly fits for the non-linear time-varying trend, the regression coefficients
of the five proxies and seasonal modes, as well as the nuisance parameters $\sigma_{\mathrm{seas}}$, $\sigma_{\mathrm{AR}}$, $\rho_{\mathrm{AR}}$, $\sigma_{\mathrm{trend}}$;
uncertainties in the nuisance parameters and regression coefficients are formally marginalized over
when stating inference of the trend, leading to a principled propagation of uncertainties. Similarly,

uncertainties in the nuisance parameters and trend can be marginalized over when we are interested
in the regression coefficients. We calculate summary statistics and confidence intervals from 90,000
samples.

    Our DLM analysis follows Laine et al. (2014) except for two key differences: we assume uniform
priors over $[0, \infty]$ for the nuisance parameters $\sigma_{\mathrm{seas}}, \sigma_{\mathrm{AR}}, \sigma_{\mathrm{trend}}$ and a uniform prior over $[-1, 1]$ for

the nuisance parameter $\rho_{\mathrm{AR}}$, rather than the lognormal and truncated-Gaussian priors used in Laine
et al. (2014). Secondly, we do not impose an external prior on the initial value of the AR1 process as
is done in Laine et al. (2014).

### 4.3  Multi-decadal changes in ozone

Here, we present estimates of changes in ozone between 1985 and 1997, and between 1998 and

2012 (Fig. 9). This is the first time that DLM has been applied to these composite datasets, including
recently updated SWOOSH and SBUV-MER. While we focus on the DLM results, we also refer to
results using MLR given in appendix Fig. A6.

    Typically, ozone trends are reported as linear decadal percentage changes in three latitude bands in
the Southern hemisphere (60°–35°S), over the equator (20°N–20°S), and in the Northern hemisphere

(35°–60°N) with sub-periods ending and starting in December 1997 and January 1998, respectively,
as shown in Fig. 9 (appendix Fig. A6 for MLR) (WMO, 2014; Tummon et al., 2015; Harris et al.,
2015). However, it does not make sense to provide a linear trend estimate for the non-linear DLM
background trend. Instead, in Fig. 9 we give the percentage change of ozone between the first and last
months of the sub-periods, i.e., between January 1985 and December 1997 (top row) and January

1998 and December 2012 (lower panels). Uncertainties represent the 95% confidence intervals of
the change for all 90,000 samples estimated with the DLM algorithm (shading for PF, bars for all
others). Since we do not show decadal trends for the DLM (but do for MLR in the Appendix), we
also show as dashed black lines in Fig. 9 the mean MLR-PF linear trend profiles from Appendix
Fig. A6, scaled from decadal changes to the longer, 13- and 15-year, sub-periods.



In the earlier, 1985–1997, period the DLM and MLR profiles agree well (within the DLM un-
certainty). The DLM-PF typically displays better agreement with the GOZCARDS profiles than the
others in the northern and southern mid-latitudes, but the mean profile is generally closer to that
of SBUV-MOD over the equator. Indeed, above 4 hPa, SWOOSH is typically at or outside the PF
95% confidence interval in all three bands (this is also the case with MLR). Interestingly, the SBUV-
composites are often outside the MLR-PF uncertainty range above 7 hPa at mid-latitudes in both
hemispheres; DLM uncertainties are larger and the four composites in closer agreement when trends
are analysed using DLM. This might hint that MLR is being biased by residual variance and/or un-
derestimating error bars, in contrast to DLM, as was observed in the test cases (see section A4).
Overall, the 1985–1997 DLM results are consistent with previous studies and MLR, with a signif-
icant decline in ozone above 7 hPa at all latitudes, especially at mid-latitudes, and negative but not
significant decreases at lower altitudes.

The results for the latter period, 1998–2012, show a significant positive trend in the upper strato-
sphere above 7 hPa, as expected to occur following the implementation of the Montreal Protocol.
The result is significant in every dataset analysed with DLM in both the northern and southern mid-
latitudes at at least one pressure level; for the PF composite the result is clear at multiple altitudes.
We note that the MLR results are only statistically significant at southern mid-latitudes for both
SBUV-composites, and for all composites in the northern mid-latitudes at 3.2 and 4.6 hPa. There are
also statistically significant differences between the mean MLR-PF and the DLM-PF profiles over
the equator and at northern mid-latitudes; in the southern region DLM profiles for composites are
less consistent than when using MLR, but the DLM PF results are in good agreement, except at 4.6
hPa. The DLM profile shapes in the northern hemisphere are consistent with each other, with a sig-
nificantly negative trend in the lower stratosphere and a positive response in the upper stratosphere,
confirming the result of Harris et al. (2015). Interestingly, with the exception of SBUV-MOD, the
large and significant negative MLR trends seen in all composites at 7 hPa disappears when using
DLM, except in GOZCARDS. This anomaly was found in an integrated-set of seven composites by
Harris et al. (2015), though not in the multi-model mean of the same composites in Tummon et al.
(2015). These results suggest that it may be an artefact of the analysis approach rather than a real
feature and further investigation is required.

In Fig. 10, we plot the DLM moving-trends as a percentage change in ozone relative to 1998;
only the PF uncertainty is presented[2], and the MLR-PF linear trends pre-1998 and post-1997 are
given as dashed lines; as a guide the MLR uncertainties are typically smaller than the DLM (see
Fig. A6). From Fig. 10, significant disagreement at 5–10 hPa at the equator and 15–32 hPa in the
northern hemisphere is very much apparent at the altitudes where DLM and MLR trend estimates
disagree on the sign of the trend; this instability of MLR was also noted by Laine et al. (2014).

---

[2]The uncertainties presented in Fig. 10 include an uncertainty on the absolute level in addition to that of the trend, while
those presented in Fig. 9 contain only the uncertainty in the trend.



Fig. 10 also allows us to observe how the background evolves with time; from this we can see that, while SBUV-MER often displays large deviations from the group (e.g. especially at 5 and 7 hPa in all latitude bands), the PF results are almost always smoothly varying and generally monotonic to/from the years 1998–2002, meaning that a comparison between MLR trends and a change between fixed dates from the DLM is indeed valid (the exception possibly being at 2, 1.5 and 1 hPa over the equator

where all datasets display relatively rapid variations in the sign of the DLM gradient, though we note this is where data are more sparse, and temporal sampling can easily be biased by the large diurnal variability; even so this altitude region appears to be where MLR and DLM are most consistent).

     Figure 10 is entirely consistent with, and explains why, Harris et al. (2015) was able to show that the choice of pivot date from the piecewise linear trend using MLR on GOZCARDS, led to larger

positive trends the later the date of pivot was chosen, i.e. from 1998 to 2002, most prominently above 10 hPa in both mid-latitude bands (see Fig. 7 of Harris et al. (2015)). We see from the DLM trends in Fig. 10 that at many locations above 10 hPa the gradient is typically zero in 2002, not 1998, especially at 3, 2 and 1.5 hPa, the exact region where the biggest increase in the trend was found by Harris et al. (2015); southern mid-latitude ozone at 1.5 hPa actually appears to start increasing a little

later, perhaps in 2004. These results are consistent between all the composites analysed, including PF.

     It is interesting to note that the two 1998–2012 mid-latitude PF profiles in Fig. 9, while determined independently of each other, display remarkably similar shapes in the DLM analysis, suggesting a symmetry in the stratospheric driving of ozone changes over this period and, indeed, a similar hemi-

spheric recovery following the Montreal Protocol. In contrast, the lower stratospheric mean-profile changes from MLR (dashed black lines in Fig. 9) are not similar, with a generally (and sometimes statistically significant) positive trend in the northern hemisphere and (an almost significant) negative trend in the southern mid-latitudes.

     We propose that the profiles determined by the DLM-PF are likely to be a better representation

of the change in stratospheric ozone than previous estimates. We base this conclusion upon: (i) that the PF was successful in identifying and correcting most known artefacts in the ozone composites, (ii) the DLM performed better than the MLR in the artificial ozone timeseries test cases, and (iii) the DLM-PF outperformed both MLR-PF and DLM of all the 'artefact-damaged' artificial timeseries. The consistency of independent northern and southern mid-latitude DLM profiles for both periods

would suggest that additional explanation for why the different hemispheres should evolve in different ways is not required (WMO, 2014). However, this also means that further investigation into why MLR and DLM trend estimates can differ so substantially is needed.

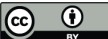


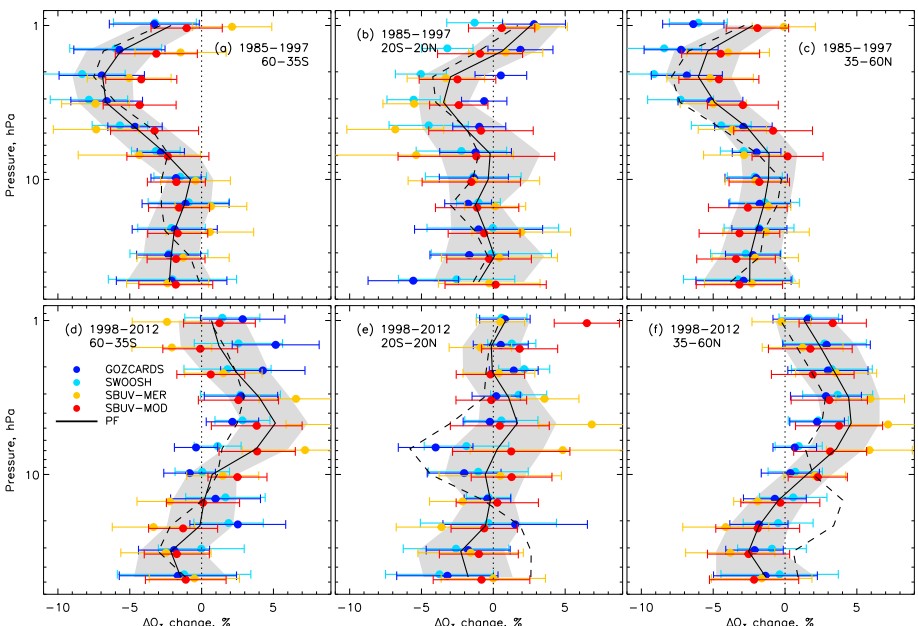

Figure 9: The percentage change in ozone from dynamical linear modelling (DLM) between 1985 and 1997 (upper row) and 1998 and 2012 (lower), over 60°S–35°S (left), 20°S–20°N (middle), and 35°N–60°N (right). GOZCARDS, SWOOSH, SBUV-MER, and SBUV-MOD are shown with error bars representing 95% confidence intervals; for the PF (black) shading represents uncertainties. The mean linear trend estimate from multiple linear regression (MLR) for the PF is given as a black dashed line (no uncertainties), and is the scaled version of the MLR-PF decadal trend shown in Fig. A6.

## 5 Conclusions

We have presented a novel approach, using a particle filter, to account for differences in ozone composites that are constructed in different ways and with observations from different sources. The need for better approaches to combine ozone composites has been raised in recent years as an issue needing resolution (e.g. Tummon et al. (2015); Harris et al. (2015)). Harris et al. (2015) stated that it is not currently possible to make definite assumptions about the best way to combine data and in what way, especially when considering multiple composites that use similar, or identical, underlying datasets. Hassler et al. (2014) noted that the key to good estimates of long-term trends is the combination of high-quality measurements and multiple instruments. Our method both requires and benefits from the availability of both. Hassler et al. (2014) further state that the consideration of uncertainties and artefacts is essential, especially when the trends are small compared to the



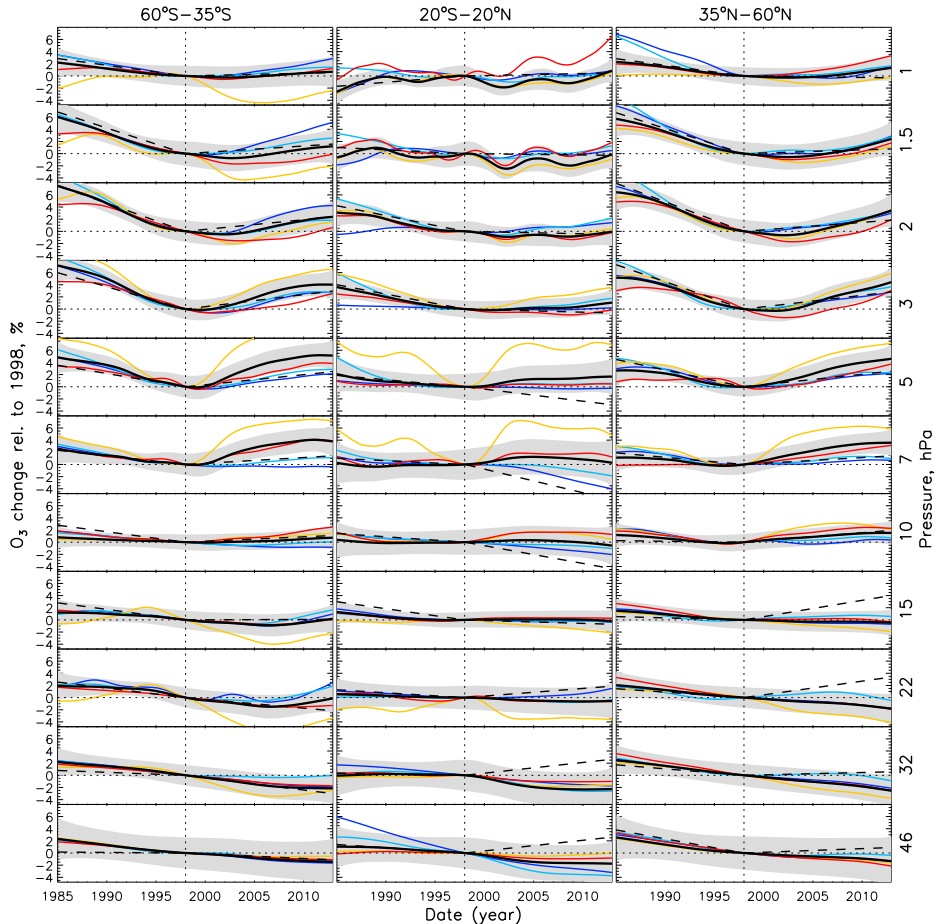

Figure 10: The percentage change in ozone (left axis) relative to 1998 (vertical dashed line; horizontal zero-line) for the integrated latitude-bands 60°S–35°S (left), 20°S–20°N (middle), and 35°N–60°N (right) and pressure levels from 1 hPa (top; right axis) to 46 hPa (bottom). Only the mean trend lines are shown for GOZCARDS, SWOOSH, SBUV-MER, and SBUV-MOD; PF is shown in black with shading representing the 95% confidence interval. The MLR trend estimates for the period pre- and post- January 1998 are given as dashed-black lines.

large natural variability (e.g. seasonal cycle), so detailed information is needed about measurement

uncertainties, data jumps due to instrument changes, and drifts. Again, our method is specifically designed to address these concerns.

The presence of data gaps, biases between instruments and issues with sampling, noise, and differences in resolution also enhance uncertainties in trend estimates, which might lead to artificial trends





being extracted in multiple linear regression analysis. To avoid this we employed, with refinements,

dynamical linear modelling (Laine et al., 2014) and found it to be more accurate than MLR when considering test cases where all variance is understood. The combination of the particle filter with DLM shows that the problems listed above can indeed be resolved to improve estimates of ozone changes on decadal timescales.

    The results presented here are a step forward, but we do not consider the composite a definitive

and final product; there are still issues to resolve, which we extensively discuss in the appendix (section A3.4). These caveats include the concern of using the same instrument dataset more than once, even though it may be used in separate composites with different pre-processing (Harris et al., 2015). Our recommendation to resolve this problem, and as the natural next step forward, is to apply the particle filter as a method to combine as many independent datasets as possible, integrating all

the known caveats and uncertainties. This will require an additional step to the methodology outlined here in order to account for absolute bias between the datasets, but we do not consider that this will cause significant difficulties.

    From the DLM analysis, the estimated changes in ozone between 1985 and 1997, and then between 1998 and 2012, show good agreement with the shape of the ozone profiles presented by Harris

et al. (2015), where seven composite datasets were combined with various approaches to estimate errors. The PF results using DLM (and MLR) show remarkably similar profile shapes and magnitudes for the earlier period. The implication for the latter period, then, is that ozone is indeed clearly and significantly recovering in the upper stratosphere as a result of the Montreal Protocol, which has not previously been demonstrated universally with significance from observations, though Shepherd

et al. (2014) demonstrated that the recovery was indeed underway by removing dynamics that interfere with calculating trends using a model with specified dynamics. The largest uncertainty in the estimates of Harris et al. (2015) came from considering instrument drift. Since the PF has accounted for much of this uncertainty, we can be confident that our smaller uncertainties represent an improvement. Further, the PF typically rejects outliers inconsistent with other composites, or otherwise

inflates uncertainty estimates, leading to our assertion that the estimated uncertainties are probably a reasonable reflection of the uncertainty in the observations. Uncertainties on the decadal trends can be further reduced with additional regressors, in addition to a new PF composite based upon independent instrument datasets rather than the four composites we considered here.

    We will male the particle filter composite, DLM and MLR trend results available, and will provide

supporting documentation should the composite be updated, in the future. Future work should extend the latitude range and time period covered, which should lead to more robust results and an up-to-date assessment of ozone trends in the stratosphere. Following our suggestion to use all available instrument data to build a new ozone composite using particle filtering, we would further recommend that this be done in collaboration with the instrument and composite principle investigators who




have the most detailed knowledge and understanding of the datasets and can provide additional, and
essential, prior knowledge.

**Appendix A**

**A1    Additional information on SBUV-composites**

In the construction of SBUV-MER, ozone was considered in 5° daily zonal means and were used
in regressions over periods of instrument overlap to account for different variability and combine
datasets into the composite; this was also used to identify and account for biases. Specific caveats
of the SBUV-MER composite include (see also Fig. 1): (i) the NOAA–11-16 overlap is very short
so only a bias offset was applied; (ii) to avoid a propagation of non-physical NOAA-9 trends to the
earlier Nimbus-7/NOAA-11 periods, Nimbus-7 and NOAA-11 are not adjusted – this is the major
difference between the dataset in Tummon et al. (2015) and the revised dataset used here – only
NOAA-9 is adjusted between the two parts of NOAA-11 and NOAA-14 is used as a bridge to the
descending part of NOAA-11, but does not appear in the final dataset; (iii) there are large differences
in the slope and intercept between 20 and 3 hPa, especially with respect to the adjustment of NOAA–
14 to NOAA–11 during the 1997–2000 overlap; (iv) while NOAA–16 and -17 are consistent with
respect to SAGE–II instrument observations, the correction approach is not as effective for NOAA–
16 and -17 at higher pressures (lower altitudes) at latitudes away from the equator.

    In the construction of SBUV-MOD, Frith et al. (2014) looked at offsets in the total column ozone
and showed that instruments typically agreed within the stated uncertainty estimates from Monte
Carlo simulations, so no additional offsets were applied to further correct them. Kramarova et al.
(2013b) and Labow et al. (2013) had also previously shown that the SBUV total ozone agrees to
within 1% with the ground-based Brewer Dobson instrument network, lidar and ozonesondes, and
was consistent with SAGE-II and Aura/MLS satellite observations to within 5%. McPeters et al.
(2013) also state that instrument overlaps agree to within ~1% in the globally-integrated (60°N–
60°S) total ozone column (TCO), although vertical profiles from NOAA-9, -11 and -14 had the
biggest non-random differences of around 2.3% between instruments at 2 hPa, related to orbit drift,
data gaps and residual uncertainties, while NOAA–16 and -18 showed differences with standard
deviations of ~1.3%. However, despite all of this, it is clear from Fig. 7 of Frith et al. (2014) that
they were able to identify offsets in the TCO - these offsets mimic the structure of the offsets between
the SBUV-composites we show in Fig. 2c, indicating that while small in total column, they are on
the order of 5% in the vertical profile, vary in magnitude and sign throughout the atmosphere, and
potentially mask offsets in the integrated column.

    Kramarova et al. (2013b) and DeLand et al. (2012) also have shown that the 1994-2000 period is
of worse quality than earlier and later periods Frith et al. (2014); DeLand et al. (2012) recommend
that NOAA–9 should not be used, which is why NOAA–14 is used for this period in SBUV-MOD,





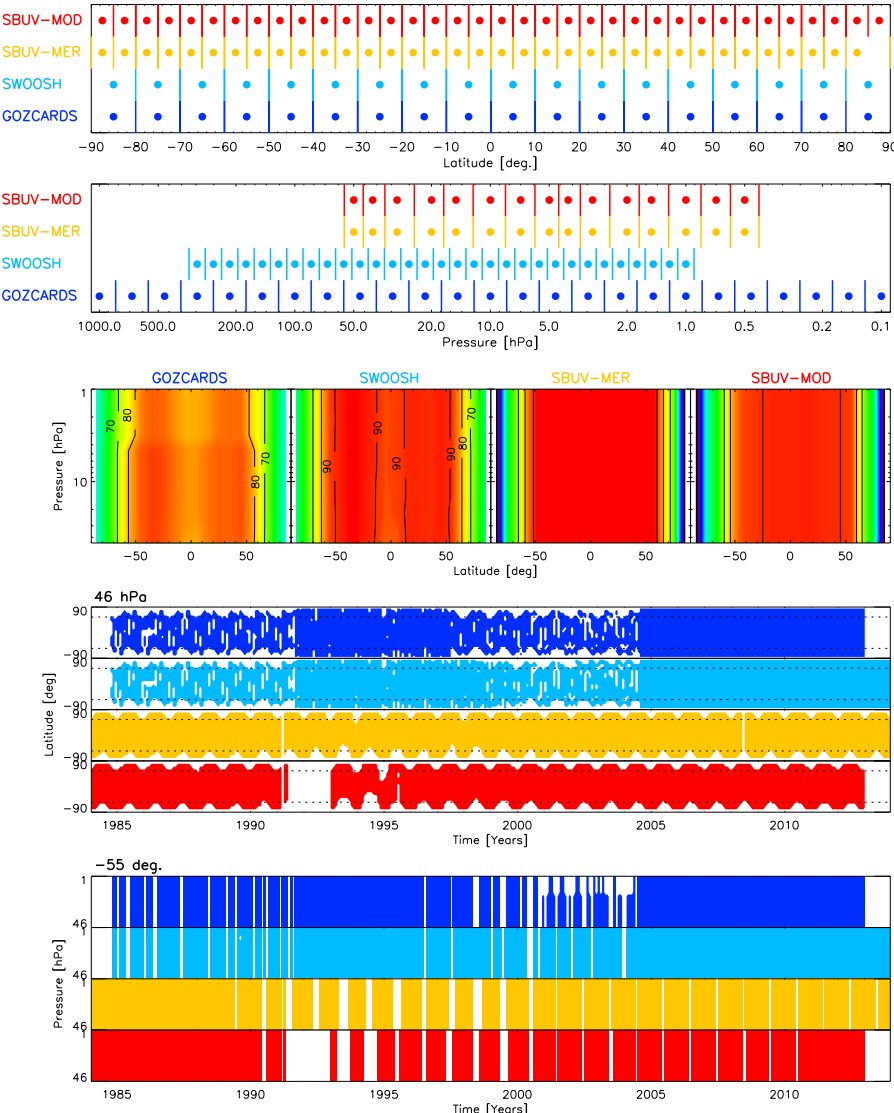

Figure A1: Visual summary of the ozone composites used here. From top to bottom: the latitudinal grid (dots represent the grid centre, lines the boundaries); the vertical grid; the percentage of months between 1985 and 2012 where data are available as a function of latitude and pressure level; the data availability as a function of latitude and time at 46 hPa; the data availability as a function of pressure level and time at 55°S. Apart from the third panel, colours are related to each of the composites: GOZCARDS (blue), SWOOSH (light blue), SBUV-MER (yellow) and SBUV-MOD (red).

although NOAA-11 drifts from 4pm to 6pm during the 1994–1995 period, for which NOAA–9 is alternatively used in SBUV-MER. A quality 'tier' for the satellites was provided in (Frith et al., 2014), which is useful in the compilation of the SBUV TCO Merged Ozone Dataset, with drifts





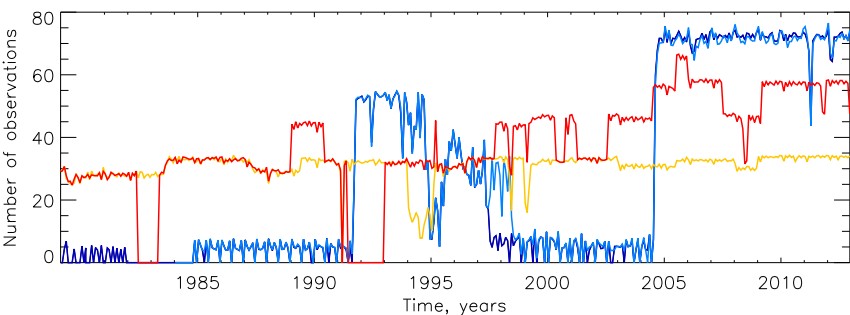

Figure A2: The square root of the number of observations at 1 hPa in each of the composites: GOZ-CARDS (blue), SWOOSH (light-blue), SBUV-MER (yellow) and SBUV-MOD (red).

tending to to cancel in NOAA–11 and –14 overlaps from 1997–2000 in TCO, but this does not reveal the profile uncertainties and drifts. The use of the priors for the PF was necessary to identify
and account for the drifts.

**A2  Additional information on the SAGE-composites**

Due to the low temporal sampling of SAGE-II (15 sunrise/sunset events per day), as opposed to the 3500 limb emission profiles per day from Aura/MLS, binning of data in GOZCARDS is done into 10° latitude averages, and datasets are connected by accounting for biases between dataset
overlaps. It should be noted that biases always exist between instruments due to calibration, spatial and temporal sampling, profile resolution, signal variability or retrieval errors. For example, Toohey et al. (2013) showed that occultation sampling errors with respect to emission measurements could reach 10–15% at high latitudes when atmospheric variability was large. The processing procedure, which occurs before data are binned into latitudes, attempts to remove outliers and impacts from
clouds or aerosols and they do not disregard data arbitrarily or attempt to fill in spatial or temporal gaps. The impact of using SAGE II v6.2 instead of v7.0 is discussed by the GOZCARDS team (Froidevaux et al., 2015), which shows very little systematic differences in number density, but leads to large differences when converted to vmr with temperature from either NCEP or MERRA (as confirmed by (Maycock et al., 2016; McLinden et al., 2009)). While small drifts of ~0.5% exist
between HALOE, SAGE II and Aura/MLS (Nair et al., 2012; Kirgis et al., 2013), SAGE II and HALOE agree to within 5% in terms of temporal stability, Nazaryan and McCormick (2005) and Hubert et al. (2015) suggest that the datasets used in GOZCARDS have good stability.

In SWOOSH, basic data pre-screening is based on published recommendations from satellite instrument teams. SAGE–II ozone screening follows the recommendations of Wang et al. (2002)
to remove aerosol contamination and poor quality retrievals; any profile containing more than 10% uncertainties between 30 and 50 km are removed. SWOOSH also applies additional screening for



profiles before November 1992 affected by the Mt. Pinatubo eruption, using information from the
NO observing channel. Offsets applied to the non-reference instrument data vary only by pressure
and latitude but not time, such that if drifts exist they may not be accounted for in SWOOSH, and
GOZCARDS.

We briefly note (and indicate in Fig. 1) technical details in the construction of the SAGE-based
composites: (i) for GOZCARDS there are no months where SAGE-II and ACE-FTS overlap in the
NH-tropics due to ACE-FTS coverage being poor; (ii) McLinden et al. (2009) noted that UARS/HALOE
and MERRA confirm that there were artefacts in SAGE-II after 30 June 2000, so these data are not
used at altitudes above 3.2 hPa; and (iii) problems with the SAGE-II azimuth gimbal in mid-2000,
and corrected by November, meant there was only a 50% duty cycle during that period, when it
already took about a month to collect data to fully cover latitudes 80°S to 80°N.

### A3 Additional information, results, and discussion on the particle filter

#### A3.1 Particle filter algorithm

#### A3.2 Success of particle filter in accounting for artefacts between composite versions

PF results in the main article uses SWOOSH data version 2.6. We originally used version 2.5 (version
2.1 was used by Tummon et al. (2015) and Harris et al. (2015)), which was updated to account for
an error which led to Aura/MLS being offset in absolute terms by one vertical level. This artefact
was clear in our original analysis, and we present an example here to show that the PF with four
composites is relatively unaffected by these types of artefacts.

In Fig. A3, we show the same results for the PF (black) and SWOOSH version 2.6 (light-blue) as
in Fig. 7a and b at 2.2 hPa and 0–10°N. In addition, we also show SWOOSH v2.5 (purple) and in
red the PF based on the same input data, but with SWOOSH v2.5 instead of v2.6 ('PF(SWv2.5)').
Prior to 2004 the SWOOSH v2.5 line is offset by ~+0.3 ppm from the zero line (i.e. relative to the
PF) and SWOOSH v2.6 in Fig. A3b. While there are small variations in the PF(SWv2.5) (red line),
it also sits close to the zero line, typically with an offset of ~+0.05 ppm and ranges between zero
and ~+0.1 ppm. We find that the PF is similarly unaffected by offsets in the previous version of
SWOOSH at other locations.

This example gives us further confidence that when multiple composites are available the PF does
a good job of accounting for artefacts that exist in only one dataset.

#### A3.3 Test of particle filter method using artificial timeseries

Given that we do not have any absolute measurements against which to test our approach we need to
demonstrate how the PF operates in ideal, known conditions, by using artificial tests cases where all





---

**Algorithm 1** Pseudo-algorithm for a particle filter applied to a single example ozone timeseries in this study

---

**Preparation:** pre-determine change in ozone between each month ('transition prior'); have (estimates of) uncertainties available; know dates of instrument changes in the composite record; know periods during which drifts or persistent problems occur and are known of; bias shift all datasets to agree for a particular time or period, preferably when confidence in the state of ozone is high [*we use the period from August 2005 to December 2012*].

**Step 1:** initialize the 'particles' by randomly sampling values uniformly for a range of values centred on the mean of the composites at the first time step [we use ±2 ppm as the range to uniformly sample over].

**Step 2:** select the observed data at the first/current time step, increase the uncertainties on the data in the composites for which an instrument change has occurred, or a known drift is occurring [*we apply a factor of 2 increase in the uncertainty in such cases*].

**Step 3:** multiply the Gaussian distributions with the mean and (possibly enhanced) standard deviation (uncertainty) using equation 2 to form the likelihood.

**Step 4:** for each particle in the prior distribution (e.g., step 1), calculate a weight equal to the value of the likelihood for the same value; sort the weights of particles in order of decreasing weight, and then normalise this sorted distribution; resample each particle by randomly sampling from a uniform distribution [with range between 0 and 1] and assigning a new value to that particle equal to the original value of the particle with a normalised weight which encompassed the value just uniformly sampled. This forms the posterior, which will be used in the next step.

**Step 5:** for each particle in the posterior (from step 4), randomly sample a value from the transition-prior and add the value to the particle to form the prior for the next month.

**Step 6:** repeat steps 2–4. Then go to step 5 and repeat until the end of the timeseries.

**Step 7:** calculate summary statistics from the posteriors for all time steps.

---

the variance is understood. With that in mind, we designed three sets of tests; we present one here and consider DLM and MLR analysis on the other two in section A4.

To create test cases, we took a real ozone timeseries, and from that estimate the regression coefficients of, solar, ENSO, volcanic aerosols, and two QBO terms using MLR (as in sectioN 4.1), and then reconstruct the ozone timeseries with these known and regressor coefficients, in addition to a realistic seasonal cycle based upon similar variability in the observations. We add a Gaussian noise term, but drop unknown residual variance. To represent instrument artefacts and drifts similar to the situation we have here with the SAGE- and SBUV- composite pairs, we produce artefact timeseries that are different between pairs, with some other differences within the pairs - these are shown in Fig. A4b as the straight lines. We add these, with different realisations of Gaussian noise for each 'instrument', to the artificial time series to produce the 'damaged' ozone timeseries shown in light-blue, blue, red and yellow in Fig. A4a. Before running the PF, we estimate uncertainties on the test cases using SVDs, and then we inform the PF when jumps (i.e. changes in the instruments, and where slopes begin and end), though not drifts, occur and we calculate the transition-prior from





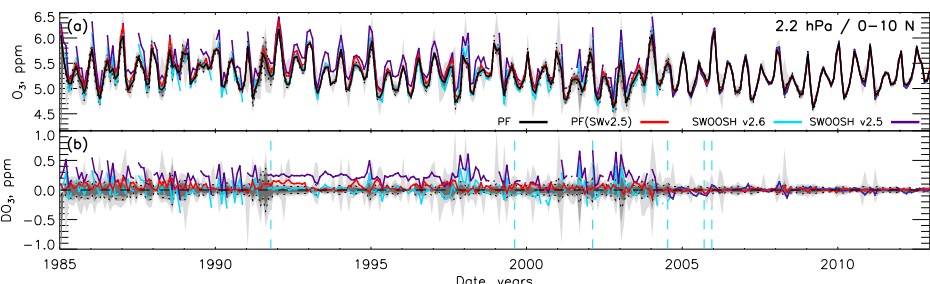

Figure A3: Ozone timeseries from 1985–2012, all bias-shifted to the mean of SWOOSH v2.6 after August 2005. (**a**) Absolute ozone at 2.2 hPa over 0–10°N from SWOOSH v2.6 (light-blue), SWOOSH v2.5 (purple), PF using SWOOSH v2.5 (red), and PF using SWOOSH v2.6 (black, with shading representing 68% (dark-grey), 95% (grey) and 99% (light grey) confidence intervals (CIs), and 2 standard deviations (dotted lines)). (**b**) As for (a), but for the difference relative to the PF(SWOOSHv2.6).

these data as for the real ozone. We then run the PF on the four 'damaged' timeseries; the result is shown in black with the 95% confidence interval in Fig. A4a. The difference of the four artificial

timeseries, relative to the original ozone timeseries (not shown), are shown in Fig. A4b.

We specifically built the artefact timeseries to provide difficulties for the PF. For example, in Fig. A4b at around month 50 all the damaged timeseries disagree with the original (undamaged, target) ozone timeseries in the same direction, to show that the PF is unable to reproduce the original ozone timeseries if none of the observations/composites correctly represent ozone during this period.

Thus, if all observations are wrong, there is nothing that can be done to resolve the issue, other than modelling using e.g. a chemistry climate model. After month 250, all the datasets are the same (i.e. there are no artefacts except the Gaussian noise that simulates instrument noise and pre-processing differences) and the PF naturally matches the artificial timeseries during this period. Prior to month 170, only one pair is either drifting or has a jump, but not both at the same time, though they are all

typically offset from the target: during this period, except when all four are different from the target (~month 50), the PF generally matches the expected ozone within the 95% confidence interval, except around months 20–30. The period between month 170 and 210 was designed to be complex, with drifts and jumps occurring within and between pairs in rapid succession. The PF, unsurprisingly, does not perform so well during this period though it doesn't generally deviate too far from the target;

between 200 and 250 it is closer to one pair, but sits between all four since there is roughly equal information and uncertainty in each of them. Throughout, when the artificial timeseries are far apart, the PF uncertainties typically increase to accommodate the higher uncertainty.



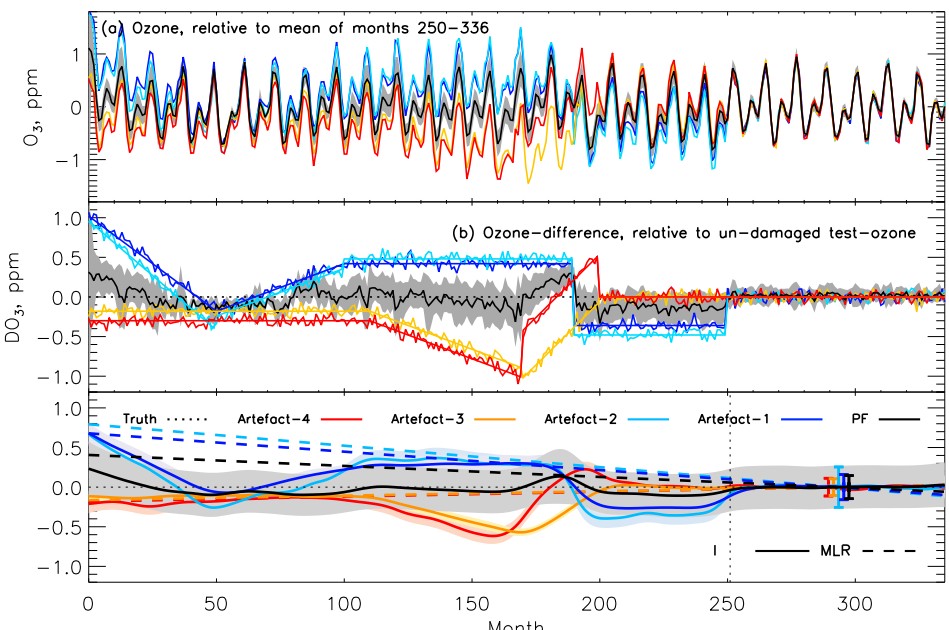

Figure A4: A test case to evaluate the performance of the particle filter. Damaged timeseries are plotted in (a) relative to the mean of months after 250 in light-blue, blue, yellow and red, and the particle filter result in black. Differences of timeseries in (a) relative to the original (test) timeseries is shown in (b); the straight coloured lines in (b) represent the artefacts applied to the original timeseries to produce the damaged ones in (a); grey and shading in (a) and (b) represent the 95% confidence intervals of the particle filter. In (c), we show the estimated trends over the full period from multiple linear regression (MLR; dashed) and the dynamical linear model in solid lines. The true trend during this period is zero (dotted line).

### A3.4    Caveats on our use of the Particle Filter

So far, we have discussed several drawbacks with the current version of the PF presented here. Here we collate and list these, and briefly discuss potential solutions for the future, where available.

1. **Vertical resolution:** This is a problem related to the different averaging kernels of the various instruments used to construct the composites - the SAGE-composites use instruments that all have higher resolution than those in the SBUV-composites. This becomes more of a problem for measurements using signals from lower altitudes, and it is clear in the case of the QBO signal being different(Bhartia et al., 2013). Kramarova et al. (2013a) recommends only using the integrated column from SBUV data below 25 hPa (16 hPa between ±20°), because although SBUV is sensitive to ozone in the troposphere and lower stratosphere, the vertical distribution





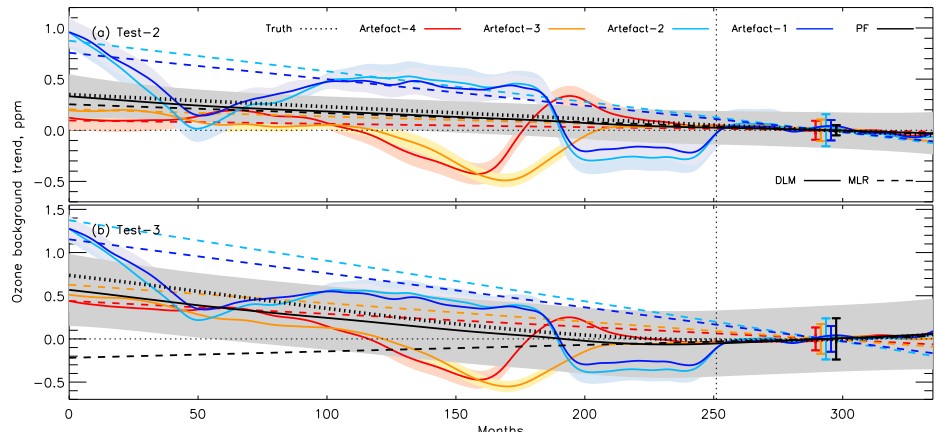

Figure A5: Similar to Fig. A4c: two additional tests cases where the only change is that the background trend in (b) is linear and in (c) non-linear, as shown with the thick-dotted black line.

of that ozone is determined by a priori constraints. Alternatively, when making direct comparisons between SBUV and other high vertical resolution instruments (e.g. Aura/MLS), Bhartia

et al. (2013) advise using the SBUV kernels to degrade the resolution of the instrument to match the vertical resolution of SBUV before comparing. However, given that some issues with resolution are already evident at 10 hPa (Fig. 7), and that there is still some useful information in the ozone observations below 25 hPa, we still consider the data relevant in this study. This issue should not represent a significant problem when MLR or DLM analysis are

performed since the two QBO regressor terms should capture much of the QBO variability. However, if one is interested in the QBO itself, then we would also recommend using the SAGE-based composites and/or datasets used to construct them (see also Kramarova et al. (2013a)). We would not endorse a solution based on de-weighting a composite relative to its vertical resolution, because then SBUV will always be at a lower weight than the SAGE-

composites and the PF will always favour the latter.

2. **Double-counting:** The use of only two pairs of composites, each built using the same underlying instrument data, resolves one of the concerns of Harris et al. (2015) about biasing our result towards the composites with the most common instrument data (e.g. five of the seven composites combined by Harris et al. (2015) used SAGE-II as a major component). However,

this leads to the problem that for periods when two of the composites are identical (i.e. not offset and with similar artefacts), the likelihood estimate may be biased in favour of that pair, which are being treated as independent datasets when indeed they are not. An example can be seen in Fig. 7b prior to 1991, where the SAGE-composites are offset from each other, but the SBUV-composites are at almost identical levels. It is fortuitous that the level of SBUV is in



close agreement with SWOOSH before 1991, and we also know that SWOOSH is offset to the SBUV-composites, and this offset remains approximately constant during this period as well as after 1991, but this may not be the case in other locations. In reality we should not treat the SBUV data as independent during this early period, but this adds further complications in making decisions about when they should be considered as independent or not. We choose not

to make this decision as this removes much of the objectivity that the PF provides. To account for this in future we recommend that the PF should be applied to the original datasets underlying the composites, each considered independently but with prior information, to construct a combined PF-composite. This would require an additional step in Algorithm 1 to estimate the offset between datasets, and to assign one dataset as a reference, but this would be a relatively

straightforward addition to the procedure.

    3. **Restricted altitude range:** We currently only consider the pressure range 47–1 hPa (~20–48 km) as we are restricted to those covered by all the composites. The GOZCARDS and SBUV-composites go higher, but observations in this region are subject to rapid diurnal changes that require good geo-location and temporal sampling, and the local time of the observations must

be taken into account. MLR trend analysis (Fig. A6) shows that the composites can display significant different long-term behaviour at 1.5 and 1 hPa, even between pairs of composites (though this is less the case using DLM in Fig. 9); this is also where diurnal variability is a serious issue as mentioned by all groups in either publications or user documentation (e.g. see McPeters et al. (2013); Davis et al. (2016) and references therein). This is an issue that is

still being investigated by the community, and we do not address it here. However, in addition to pre-screened data, it may be something that is possible to resolve with accurate transition-priors, and additional prior information, in addition to the ones we already suggest using here. Observations are also available down to 316 hPa, but there are large gradients in ozone at these levels, so even the relatively high resolution of the instruments in the SAGE-composites can

struggle to accurately resolve variability at individual layers this low down. However, many observations do exist, and so when integrating the original data using the PF (see previous point), these layers could be included, and additional prior information could also be used to account for the large ozone gradients.

    4. **Restricted latitude range:** While the composites extend to higher latitudes than 60°, at these

latitudes the need for direct or scattered sunlight leads to several months of the year where data are missing, with increasing periods of the year without observations closer to the poles. We do not attempt to fill these data without observations available. In future, we could use night-viewing instruments such as GOMOS (Kyrölä et al., 2013), to extend into higher latitudes when these data are available (i.e. after 2002), but it is not possible to do it prior to the GOMOS

measurements, except potentially through ground-based observations, though they are usually





limited to lower altitudes than the satellite observations can consider. In future we could also consider extending the PF to better estimate ozone during at least the summer seasons.

5. **Mt. Pinatubo:** Even though the example given at 10 hPa clearly indicates that the PF is able to avoid the artificial decrease in the SBUV-MER data between June 1991 and 1992, it does not perform well at 15 hPa (see also section 3.6). Frith et al. (2014) advise caution when using data in the 6–9 months following the eruption, especially for 15°S – 30°N. Thus for this altitude, when using MLR to analyse trends, we also suggest ignoring this period at 15 hPa. One solution would be to increase the prior de-weighting factor over this period, but this would be an additional subjective decision, so we prefer to flag this information instead and find a more elegant solution in the future.

Some of these caveats may be resolved with additional information from the ozone community and by using the PF to construct a composite from the original, individual instrument timeseries. Nevertheless, for the work involving composites here, we conclude that despite these issues, overall the PF performs well in estimating ozone variability. This conclusion is based upon the artificial test case target timeseries being well estimated, the results of the example real ozone timeseries presented in Fig. 7 that account for known issues, and the success in the case of the SWOOSH version changes where the PF accounts for the problems in SWOOSH in v2.5 in advance of the v2.6 release (section A3.2).

**A4    Comparison of multiple linear regression and dynamical linear modelling in estimating long-term trends**

To test the ability of MLR and DLM to estimate the background trend, we use the artificial test cases presented in appendix section A3.3 and Fig. A4a, in addition to two more with the same regressor-coefficients and noise, but with a linear and non-linear time-varying background trend (Fig. A5). The first set have a background, linear, zero-trend (Fig. A4c), the second a linear downward trend (Fig. A5a), and the third a downward-linear trend plus a non-linear curve that reaches a minimum in the latter half of the full period before increasing again (Fig. A5b); the true 'target' trends are shown in Fig. A4c and Fig. A5a and b as thick-dotted black lines. In each case we apply the particle filter to the four sets of artefact-damaged timeseries, as in Fig. A4. Therefore, we have 15 test timeseries, all fully understood. This does not represent the situation for the real ozone timeseries since in many of those cases the MLR residuals (unaccounted for variability) can typically account for ~50% of the variance. However, these tests with artificial ozone timeseries are indicative of the performance with real timeseries.

One major advantage of DLM over MLR for estimating long-term trends is that MLR requires the trend to be prescribed in advance as linear, or piece-wise linear trends (e.g. Kyrölä et al. (2013), or is expected to follow the equivalent stratospheric chlorine (EESC) curve (Newman et al., 2001). The





shape of the EESC, which follows CFC stratospheric loading that peaked in the mid-to-late 1990s, impacts more on the sensitivity of the MLR analysis than the period length does when calculating decadal trends WMO (2011). The main problem in assuming an EESC shape is that the timing of chlorine minimum is location dependent with, e.g., higher latitudes lagging those closer to the

equator since it takes time for chlorine changes to reach different regions. Therefore, fixing the decline date may lead to misleading estimates (Harris et al., 2015). The use of the DLM allows this issue to be circumvented to some degree by not fixing the background trend or an inversion date (Laine et al., 2014) and allowing it to vary with time, though this still does not necessarily separate EESC from dynamical changes related to, e.g., changes in the BDC (Polvani et al., 2011; Harris

et al., 2015).

In Fig. A4c, we plot the MLR (dashed) and DLM (solid) trend results. In this example the true long-term trend is zero (dotted black line). The only result that is able to stay within two standard deviations of the 'truth' is the DLM of the PF, and usually it is within one standard deviation. The MLR of the PF shows a significant downward trend, and naturally one would not expect the MLR

of the damaged timeseries to estimate an accurate result. What is interesting to observe is that the DLM accurately extracts the damaged background trend as well, which might be useful in future studies to further assess anomalous behaviour in the composites by interpreting the behaviour of the DLM results from each composite. The two tests with the linear and non-linear background trends (Fig. A5) lead to essentially the same conclusions. A more thorough assessment of the DLM with

respect to MLR will be made in a forthcoming publication.

In summary, our tests suggest that when estimating the long-term trend, the use of the PF to correct data, together with the DLM, is more successful and accurate than using MLR or DLM on uncorrected timeseries. Therefore, we would recommend using the PF combined together with the DLM for the analysis of long-term trends in ozone, as outlined in this study.

*Acknowledgements.* We thank S. Frith, J. Wild and L. Froidevaux for detailed comments on the composite datasets and general comments on the manuscript. We also thank the GOZCARDS, SWOOSH, SBUV-MOD and SBUV-Merged Cohesive composite teams for use of their data. W. T. Ball and E. V. Rozanov were funded by Swiss National Science Foundation (SNSF) grants 200020_163206 (SIMA). Fiona Tummon was funded by SNSF grant 20F121_138017. GOZCARDS ozone data can be found at https://gozcards.jpl.nasa.gov/.

SWOOSH ozone data can be found at http://www.esrl.noaa.gov/csd/groups/csd8/swoosh/. SBUV ozone data can be found at http://acd-ext.gsfc.nasa.gov/Data_services/merged/. The PF composite will be made available following the review process.





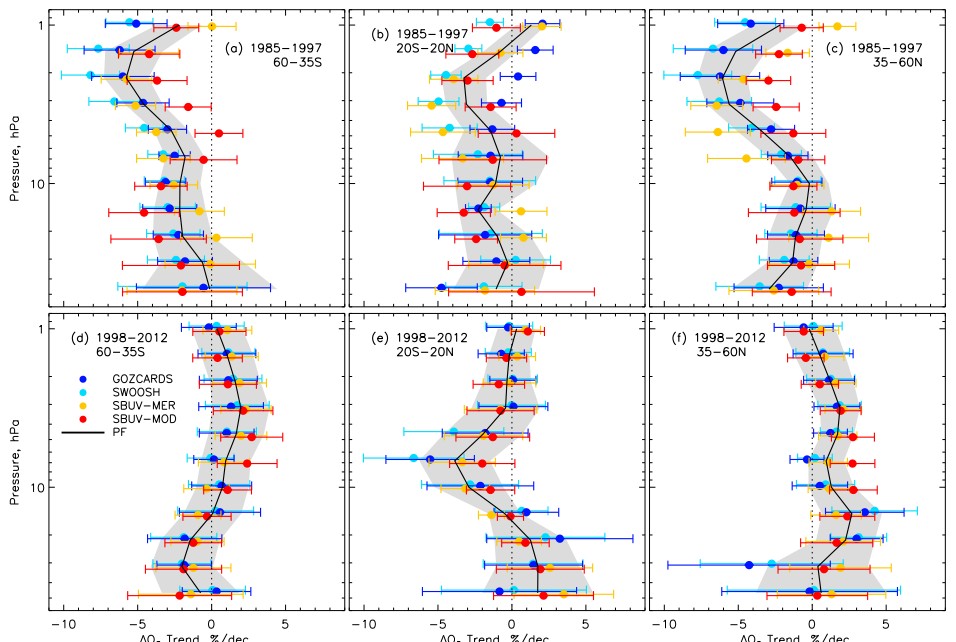

Figure A6: The decadal trend in ozone from multiple linear regression (MLR) between 1985 and 1997 (upper row) and 1998 and 2012 (lower), over 60°S–35°S (left), 20°S–20°N (middle), and 35°N–60°N (right). GOZCARDS, SWOOSH, SBUV-MER, and SBUV-MOD are shown with 95% confidence intervals; PF is shown in black with shading representing 95% confidence intervals.

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
