# Peer review of "Reconciling differences in stratospheric ozone composites"

_Atmospheric Chemistry and Physics, 2017_

## Referee Comment (RC1) · M. Laine (Referee) · 20 Apr 2017

The article deals with the important topic of combining retrieval or composites from several satellite instruments. It nicely illustrates the difficulties in trend estimation with multiple composites having different characteristics. It provides a valid method for constructing a combination of ozone composites from different sources. This a quite general topic. To be able to detect changes in environmental processes, long time series are needed and this in turn leads to need for sound data fusion techniques. Another general topic is that trend analysis crucially depends on realistic uncertainty estimates.

The composites are merged with what the authors call particle filter method. In addition they produce uncertainty estimates for individual composite observation by SVD method. The merged data sets are analysed for trends by using dynamic linear model

(DLM) and multiple linear regression (MLR). The article compares the methods and gives recommendations.

As for the recommendations, I agree that DLM is the correct basic framework for analysing time series. It can be seen as a hierarchical Bayesian model or in classical statistical settings, if needed. It has the MLR as a special case, so one can study the need for a "smooth" trend analysis, instead of linear trend. In addition, change points do not have to be prescribed, but their locations can be estimated from the data, and all assumptions can be studies statistically.

I have a couple of general comments about the presentation of the methodology.

**The PF method is presented like a model, but in fact it is a numerical algorithm**

For example
line 17: "Particle filtering and DLM",
line 20: "The particle filter results",
line 779: "using a particle filter",
line 804 "the particle filter as a method".

In my opinion, the distinction between a model for data and a numerical algorithm should be made more clear. You should first describe the model (your dynamical mixture-Gaussian model as a Bayesian hierarchical model) behind the data merge and then the numerical Monte Carlo filtering algorithm (PF/SIR) for actually estimating the merged data set.

PF (or SIR) is a numerical method of computing a certain Monte Carlo estimate of a posterior (predictive) distribution in a dynamical model. You propagate an ensemble ensemble of possible model states (time series realizations) by a model (here the assumed month-to-month seasonal change and known deficiencies) to produce prior ensemble for the next state, which is then weighted by the likelihood function defined by the observed satellite composites. This will form a sample of the posterior uncer-

tainty of the merged series given the observation up to the current time point. In effect this is a non-linear, non-Gaussian generalization of a Kalman filter.

You could contrast this to DLM or MLR "methods". DLM (and MLR, too) is a model for the processes and the system generating the observations (see below for a general state space description. DLM is a structural state space model that constructs a time series from basic building blocks, like trend, seasonality and proxies. For DLM one can use Kalman filter and smoother as an estimation algorithms. For MLR you can use the least squares algorithm for estimation, but other algorithms are available, as well.

**SVD for uncertainty estimates**

A similar comment is valid for the SVD "method" for construction of uncertainty estimates for the individual composites. SVD is an algorithm for a certain matrix decomposition. For the uncertainty analysis, you will have a some kind of model based on principle components and then you use the SVD algorithm for estimating the components. Is there any references the "SVD" approach used? I think the approach would need more motivation. You could write a model for the sources of uncertainties for each composite, having a common source and other sources that might be instrument specific. Then you could estimate these by principle components. As an example, a model for composite $d_i$ would be $d_i = P_i T = p_{1i}T_1 + p_{2i}T_2 + p_{3i}T_3 + p_{4i}T_4$, where $T$ are the principle components and $p$ are the corresponding loadings. Then use it to build a model for variance components of a composite $d_i$, as $\mathrm{var}(d_i) = ...$, that would include the composite uncertainty as one of the components.

**Filter vs. smoother**

You should motivate why "filtering" is adequate for the data merge and no "smoothing" is needed. A filter calculates $p(y_t|\{d_t\})$ for each $t = 1 : T$, but not $p(y_t|\{d_{1:T}\})$ nor the joint distribution $p(y_{1:T}|\{d_{1:T}\})$. The latter are what are estimated by Kalman smoother in DLM calculations for a linear state space model.

Additional question: could PF be replaced by suitable weighted average of the composites, that just takes into account the prior information about problems in the individual series? In DLM and MLR you will need to assume Gaussian uncertainty, so the PF results need to be summarized as mean and standard deviation. What are the benefits of PF over some simpler (non-Monte Carlo) averaging method?

**About MCMC**

I would like to see some MCMC results for the DLM analysis. You are using uniform priors for the variance parameters (line 689). Do these parameter identify, especially, if you assume unconstrained smoothness for the trend? How do the AR parameters identify? You use uniform $[-1, 1]$ for the AR parameter, but do you consider negative autoregression as a realistic model for an ozone observation time series? You could include some plots of the posterior distributions.

**General state space model approach**

I suggest that you describe the merge and trend analyses as a general hierarchical state space model. In both merging the data and in the DLM analysis you are dealing with a dynamical state space model. A general framework to describe the statistical model is by a hierarchical description, with a process model for the model state dynamics, a parameter model for model (nuisance) parameters and a data model for the likelihood. The Bayes formula would provide the posterior estimate from the individual conditional components as (see [1,2,3]):

[process, parameters|data] $\propto$ [data|process,parameters][process|parameters][parameters]

Filtering and smoothing algorithms can be used to estimate various marginal and conditional posterior distributions. The nuisance parameter could be integrated out by MCMC, for example.

For ozone data merge the process model includes the month-to-month variability and external events like volcanos, trends etc. The observation processes could describe

the instrument effects. Lastly, there is the prior distributions for model parameters. The whole will in effect be a hierarchical Bayesian model to describe and estimate the state together with the parameters. This could provide a common framework for both merging and analysing.

[1 ] L. M. Berliner. Physical-statistical modeling in geophysics. Journal of Geophysical Research: Atmospheres, 108(D24):8776, 2003. doi: 10.1029/2002JD002865.

[2 ] C. K. Wikle and M. B. Hooten. A general science-based framework for dynamical spatio-temporal models. TEST, 19(3):417–451, 2010. doi: 10.1007/s11749-010-0209-z.

[3 ] N. Cressie and C. K. Wikle. Statistics for Spatio-Temporal Data. Wiley, 2011.

**Other comments**

line 385, equation (2): I do not see how the parameters $\gamma$ and $\beta$ give rise to bimodality for an individual composite as the mean is the same $d_t^c$ for both modes. It probably will make the tails of the likelihood heavier than for a standard Gaussian likelihood.

line 460: The PF distribution is said not to be Gaussian but in DLM and MLR you need Gaussian uncertainty. Is this a problem for the trend analysis?

line 801: "using the same instrument dataset more than once". The transition prior is inferred from the same observations that are used in the model, so the data is used twice. Also, the uncertainty is inferred from the same data by SVD. Maybe this is ok here, but it violates the Bayesian assumptions.

Can you elaborate more the claim that PF method can resolve the problems in data merging? Do you claim that PF is capable to extract the background truth behind

different biased estimates. Or does it just make the "error bars" larger, so that the trend analysis is not affected by instrument artefacts?

I agree that construction of a merged data set is of interest in itself. For trend analysis one could start from individual observations. You could discuss the possibility of a general data fusion approach that assimilates all the different composites or individual retrievals to a common time series model. You might still be able to use linear model, but with carefully designed (linear) observation operator, that would account for the instrument artefacts. Or use some non-linear generalization of DLM.

**Conclusion**

I can recommend the article to be published, if the author formulate the modelling approach for merging and uncertainty estimation a little more consistently, motivate the adequacy of the filter in the data merge and the use of SVD for the uncertainty variance components, and describe the MCMC results for DLM.

---

## Referee Comment (RC2) · D. Hubert (Referee) · 14 May 2017

**1  General comments**

**1.1  Relevance**

It is an inherently challenging task to estimate the magnitude and the patterns of the uncertainties in a data record constructed from measurements by various instruments. Yet succeeding in this endeavour is crucial when the strength of the geophysical signal of interest is similar to the level of uncertainty in observations. The determination of trends in stratospheric ozone observations is a textbook example of this challenge, one that has caused a lively debate within the community over the past few years.

[Figure]

Discussions revolved not only around how to quantify the uncertainties in the data sets and how to incorporate these in the analyses, but also around the regression methods used in time series analysis. This paper proposes alternative methods that are of immediate interest to the ozone community. And to the broader atmospheric community as well, since inferences of climate parameters are often based on (a set of) composite observational data records.

**1.2 Methods**

The authors adopted a series of techniques to exploit the information contained in four merged data sets to arrive at the probability density function of ozone at each time step, which is then analysed for long-term background trends. To my knowledge, each of these techniques has only rarely or not been applied to ozone data so the results of each sub-analysis is interesting in its own right.

First of all, the singular value decomposition analysis results appear to improve existing estimates of uncertainty in the four ozone composites. I encourage the authors to release these uncertainty estimates, since these are relevant to users of the composites.

Then, this information is used by a particle filter, a sequential Monte Carlo algorithm, together with the transition probability from one month to another extracted from the data. Ball et al. build a convincing case that the particle filter is indeed fairly robust against known or –in some cases– previously unidentified deficiencies in individual composites. Various illustrations show the ability of the particle filter to either adjust for a sudden jump and/or temporary drift, or, to inflate the uncertainty when the information contained in the ensemble is insufficient. The algorithm also unravelled issues not known beforehand, which shows its potential as a tool to improve the considered data sets.

The paper ends by a discussion of the limitations of traditional linear methods in

analysing ozone time series and the authors apply the non-linear method proposed by Laine et al. Doing so brings the lower stratospheric trends from different hemispheres in better agreement and the results seem more stable in the tropical middle stratosphere as well.

**1.3 Presentation**

There is a vast amount of information in this paper. Perhaps a bit too lengthy in some places, so some trimming here and there (mainly in Section 3) could improve the paper. But overall the authors succeeded in delivering an easy to read, detailed yet fairly concise account of several novel methods and results. I also appreciated the effort they have put into the graphics and the appendix, which contains an abundance of useful and instructive supplementary material.

**1.4 Conclusions**

I commend the authors for their research and this paper, which is highly relevant to a broad atmospheric research community. It surely fits the scope of ACP and I recommend publication once the minor comments below are addressed.

**2 Minor comments**

- l.142, p.5: There is still a few % diurnal component between 1-5 hPa which will alias into the long-term trend for uncorrected measurements from instruments on drifting orbits. Please clarify that diurnal variations are not entirely avoided, only those with largest magnitude above the stratopause.

- l.172-177, p.7: I doubt an uninformed reader will grasp the message in "[...]

SBUV-MER considers only one data set at a time [. . .], while SBUV-MOD averages overlapping data to combine them [. . .]". A slightly more verbose description of the SBUV-MER merging approach may make the difference with that of SBUV-MOD clearer.

- l.222-223, p.9: Not all occultation instruments retrieve O3 number density on altitude levels, only the UV-VIS instruments do so (SAGE-II). IR occultation missions (HALOE, ACE-FTS) retrieve O3 volume mixing ratio, some even on pressure levels. Please correct this statement.

- l.352-362, p.13: What's the rationale for the factor 2 increase, and how sensitive are the Particle Filter results to this choice? Over what timescale is the uncertainty expanded? Just the month following the change or is it smeared out over a number of months?

- l.368-369, p.13: How prohibitive is the assumption of uncorrelated measurement errors for the joint-likelihood function? The bottom row of Fig. 4 clearly demonstrates the correlation of the uncertainties in time and between composites.

- l.377-379, p.14: Assuming that $\beta = 10\%$ of the observations need a blow-up of their uncertainty by $\gamma = 100$ is quite harsh. I would expect smaller values for $\beta$ and especially for $\gamma$, whose effect would be to reduce the tails of likelihood. But perhaps your choice is more of a worst-case scenario? How sensitive is the Particle Filter outcome to the choice of $\gamma$ and $\beta$?

- l.403-413, p.15: It should be mentioned here that no transitions were used when an instrument changed. This relevant information is now hidden in the caption of Fig. 5.

- l.440, p.15: I had to wait for 43 lines to find out how large N is. I would mention this already from the start and come back to its motivation at the end of the section.

[Figure]

- l.441-442, p.17: You praised the benefit of using a non-Gaussian likelihood (sum of Gaussians) in Sect. 3.2, so it is confusing to read about Gaussian composite likelihoods here. My eye cannot distinguish the likelihoods in Fig. 6 from Gaussian distributions (which also touches on the topic in a previous comment about $\beta$ and $\gamma$), the former should be more heavy-tailed. I would just drop the "as Gaussian distributions".

- l.45-462, p.17: I found these couple of phrases (ending with "Fig. 6c.") of little value for the paper, as they essentially give a technical explanation of the resampling procedure. Or did I miss something?

- Fig. 6, p.18: You may want to point somewhere in the paper to the outlying GOZCARDS likelihood in panel (j) which has a clear impact on the 99% credible region. I found this a nice illustration of the multi-modal joint likelihood.

- l.536-539, p.20: Is the transition prior of PF(SAGE) bootstrapped from the transitions of the two SAGE-composites rather than from the four composites?

- l.553-554, p.20: This phrase is strange, perhaps part of it is missing. How can local time of equator crossings be near-polar to attain near-global coverage?

- l.589, p.23: OSIRIS is a limb-viewing instrument, so should not be mentioned here.

- l.606, p.23 (and elsewhere): The notion of "trend" carries various meanings in the community. Personally, I preserve "trend" for any long-term component that can not be attributed to known atmospheric processes or to known measurement artefacts. I advocate the phrasing "drift" or "artificial trend" in the latter case, which is much less confusing than blending it in with actual geophysical signal.

- Sect. 4.3: How did you go from the time series in 10 degree latitude zones to regression results over 30 degree wider latitude belts? Average the time series,

then regress? Please explain this briefly in the manuscript.

- l.752-754, p.28: Do you (or Laine et al) have an explanation for this instability? If you don't, perhaps mention that this deserves further investigation. This feature is striking and should be better understood.

- Fig. A2, p.34: Specify the latitude range unless the Figure is for the entire data set at 1 hPa.

- l.894, p.34: Add units to "small drifts of 0.5%". I know many people refer to 0.5% per year (or 5% per decade) as small, but they actually mean small compared to the stability of the data records. It is definitely not small compared to the actual trend being targeted, so this is a very unhappy phrasing in my opinion. Same comment for "[...] SAGE and HALOE agree to within 5% in terms [...]", what is the unit?

- l.895, p.35: Hubert et al. (2016) is the first official report of a significant drift of 5% per decade of HALOE relative to sonde and lidar. Previous studies are consistent with this negative drift, but the results were not significant. I suggest to nuance your statement slightly: "[...], Nazaryan and McCormick (2005) and Hubert et al. (2016) suggest that MOST datasets used in GOZCARDS have good stability."

- Algorithm 1 (Step 3), p.36: See earlier comment, the composite likelihoods are not Gaussian according to Eq. 3.

- Algorithm 1 (Step 4), p.36: See earlier comment, isn't this just a technical description of implementation? I would summarise this to one phrase.

- l.950, p.37: The word "original" is somewhat ambiguous here; it could mean the real, observed time series or the fit (with/without Gaussian noise?) to that time series. I find the phrasing "undamaged" time series better here (also used as label in Fig. A4b).

- l.962, p.38: I don't see months 20-30 as a second exception, they are just a result of the offset of all four composites in month 50.

- l.973-974, p.39: This phrase confused me, do you mean that the vertical resolution of SBUV degrades at lower altitudes? Perhaps you canted to say "This difference in vertical resolution becomes more important at lower altitudes."?

**3 Technical corrections**

- l.125, p.4: The Penckwitt et al. paper was (and will likely) not (be) published. Please double-check this with G. Bodeker, one of the authors. Alternatively, Tummon et al. (2015) probably remains the best reference for this data set, as it has a concise summary of the merging method, satellite instruments and data versions.

- l.181-182, p.8: Not sure where the "this" refers to in "[. . .]; this describes the updated version [. . .]".

- l.207, p.8: Reference to Fig. 2b, should be to Fig. 2c.

- Fig.3, p.9: It is hard to discern blue from black markers/lines in print. Perhaps this Figure will benefit from a deviation of the colour scheme used in the rest of the paper.

- l.233, p.10: Replace "SAGE-II-based instruments" by "SAGE-II".

- l.283-286, p.11: The section references are incorrect.

- l.378, p.14: Smaller values of $\beta$ encode more faith in individual observations rather than less, no?

- l.390, p.15: "compositeS".

- l.435, p.17: Remove "(Algorithm 1)" following "the preparation step".

- l.672, p.25: "deseasonAlised".

- l.730-731, p.27: A "negative decrease" is, strictly speaking, an increase. Could be replaced by "[...], and insignificant decreases at [...]".

- l.873, p.34: The colon messes up the citation. Which of Frith and DeLand (perhaps both) recommends that NOAA-9 should not be used?

- l.878, p.34: Remove duplicate "to" in "[...] tending to to cancel [...]".

- l.895, p.35: Hubert et al. (2015) became Hubert et al. (2016) in the meantime.

- l.914, p.35: Add a reference to Algorithm 1 on the next page, so this section is not empty.

- l.937, p.36: Typo in "sectioN".

- Fig. A4, p.38: Fix the legend label for "DLM" in the bottom panel.

- Fig. A5, p.39: Fix labels in caption, should be (a) and (b) instead of (b) and (c).

- l.975, p.39: Add a space before reference to Bhartia et al.

- l.1168-1174, p.45: Update reference to AMT version of manuscript.

---

## Author Comment (AC1) · 28 Jul 2017

**Atmos. Chem. Phys. Discuss., doi:10.5194/acp-2017-142-RC2, 2017**

**General response**

We thank both referees for their thorough consideration and constructive feedback. As a result of the review process, we have made a significant effort to improve semantics regarding methods, models and algorithms. In the revised version of the manuscript we have replaced the particle filtering method by a Hamiltonian Monte Carlo (HMC) approach to sample the full posterior distribution, conditioned on the full data vector as is required, rather than just the data up to time t, as in the particle filter. HMC is well documented in the literature, and as such, the length section on the particle filter has been significantly reduced. We re-did the sampling algorithm from scratch, re-ran everything and re-made all relevant plots. In practice, we found that the new results are broadly similar to the particle filtering results and none of the key findings are changed. We now refer to the composite constructed using Gaussian-mixture likelihood and transition prior, with SVD uncertainty estimates, as the BAyeSian Integrated and Consolidated (BASIC) composite.

We note that there are some differences that you should be made aware of compared to the previous version. These include:
- the time-dependent error bars are much tighter, and much closer to Gaussian than before; this is good because the DLM analysis will better represent the data with Gaussian errors included;
- the problem we found (in only a few limited regions) following Pinatubo has gone and BASIC performs well during this period, given the data supplied to the process;
- there is a longer section in the appendix that encompasses requests for information on the Gaussian-mixture likelihood construction (in BASIC), and the parameter estimation (in the DLM);
- Northern and southern hemispheres in the profiles were actually the wrong way around; given the symmetry between hemispheres, the conclusions do not change.

We reply to all comments below, with referee comments in black, and our responses in blue.

**Marko Laine (Referee)**
**Received and published on 20 April 2017**

*The PF method is presented like a model, but in fact it is a numerical algorithm*

For example
line 17: "Particle filtering and DLM",
line 20: "The particle filter results",
line 779: "using a particle filter",
line 804 "the particle filter as a method".

In my opinion, the distinction between a model for data and a numerical algorithm should be made more clear. You should first describe the model (your dynamical mixture-Gaussian model as a Bayesian hierarchical model) behind the data merge and then the numerical Monte Carlo filtering algorithm (PF/SIR) for actually estimating the merged data set.

PF (or SIR) is a numerical method of computing a certain Monte Carlo estimate of a posterior (predictive) distribution in a dynamical model. You propagate an ensemble ensemble of possible

model states (time series realizations) by a model (here the assumed month-to-month seasonal change and known deficiencies) to produce prior ensemble for the next state, which is then weighted by the likelihood function defined by the observed satellite composites. This will form a sample of the posterior uncertainty of the merged series given the observation up to the current time point. In effect this is a non-linear, non-Gaussian generalization of a Kalman filter.

You could contrast this to DLM or MLR "methods". DLM (and MLR, too) is a model for the processes and the system generating the observations (see below for a general state space description. DLM is a structural state space model that constructs a time series from basic building blocks, like trend, seasonality and proxies. For DLM one can use Kalman filter and smoother as an estimation algorithms. For MLR you can use the least squares algorithm for estimation, but other algorithms are available, as well.

We agree that the semantics regarding methods, models and algorithms needed cleaning up. In the updated version we refer to the composite constructed using Gaussian-mixture likelihood and transition prior, with SVD uncertainty estimates, as the BAyeSian Integrated and Consolidated (BASIC) composite, and refer elsewhere to specific methods and algorithms appropriately.

**SVD for uncertainty estimates**

A similar comment is valid for the SVD "method" for construction of uncertainty estimates for the individual composites. SVD is an algorithm for a certain matrix decomposition. For the uncertainty analysis, you will have a some kind of model based on principle components and then you use the SVD algorithm for estimating the components. Is there any references the "SVD" approach used? I think the approach would need more motivation. You could write a model for the sources of uncertainties for each composite, having a common source and other sources that might be instrument specific. Then you could estimate these by principle components. As an example, a model for composite di would be di = PiT = p1iT1 + p2iT2 + p3iT3 + p4iT4, where T are the principle components and p are the corresponding loadings. Then use it to build a model for variance components of a composite d_i, as var(di) = ..., that would include the composite uncertainty as one of the components.

We have updated the discussion of the uncertainty estimation in Section 3.1 to give more clarity about the role of the SVD (essentially as a numerical method to implement PCA) and also to more fully explain our heuristic error estimation. There are references to use of SVD, but none we are aware of in the form we have put forward here. There are no references for this method itself, but we have done empirical tests to show that it produces sensible results for reasonably discrepant data-sets such as those being analysed here. That said, we explicitly state that more work is needed on this aspect of the overall data analysis problem and we fully expect to attempt this in future papers.

**Filter vs. smoother**

You should motivate why "filtering" is adequate for the data merge and no "smoothing" is needed. A filter calculates p(yt |{dt}) for each t = 1 : T, but not p(yt |{d1:T }) nor the joint distribution p(y1:T |{d1:T }). The latter are what are estimated by Kalman smoother in DLM calculations for a linear state space model.

Additional question: could PF be replaced by suitable weighted average of the composites, that just takes into account the prior information about problems in the individual series? In DLM and MLR

you will need to assume Gaussian uncertainty, so the PF results need to be summarized as mean and standard deviation.

What are the benefits of PF over some simpler (non-Monte Carlo) averaging method?

Thanks for pointing this out. You are correct that the particle filtering algorithm samples from the posterior distribution of the true time series conditioned on the data "up to that point", rather than the full data vector, and conditioning on the full data vector would require a subsequent smoothing step. The smoothing step comes with some considerable technical difficulties - to get around this whole issue, in the new version we have abandoned the particle filtering method completely and resorted to Hamiltonian Monte Carlo (HMC) sampling to sample the full posterior distribution, conditioned on the full data vector as is required. HMC is well suited to ultra-high dimensional sampling problems and is well documented in the literature. We re-coded the sampling algorithm from scratch, re-ran everything and re-made all relevant plots. In practice, we found that the new results (now correctly conditioned on the full data vector) are broadly similar to the particle filtering results and none of the key findings are changed. Nonetheless we thank you again for pointing this out and the new approach is now correct and more robust.

Regarding to what extent our method is akin to performing a weighted-average of the composites: for sure, some of the data-artefacts will be reduced/removed by taking an inverse-variance weighted mean, and for a lot less effort. However, use of the (fat-tailed) Gaussian-mixture likelihood combined with the month-to-month transition prior allows our approach to identify where certain data are corrupted without a priori knowledge of specific issues — these (many) cases cannot be captured by simply averaging. We also provide non-Gaussian uncertainties; it's true that DLM/MLR assumes Gaussian errors, but we encourage extension of these tools to allow for non-Gaussian uncertainties and/or marginalization over a full systematics model - this is a first step on a long road to a more principled approach to trend analysis from ozone data.

**About MCMC**

I would like to see some MCMC results for the DLM analysis. You are using uniform priors for the variance parameters (line 689). Do these parameter identify, especially, if you assume unconstrained smoothness for the trend?

How do the AR parameters identify?

You use uniform [−1, 1] for the AR parameter, but do you consider negative autoregression as a realistic model for an ozone observation time series?

You could include some plots of the posterior distributions.

Thanks for raising these issues. Over the course of the work we experimented with different prior assumptions for the DLM. We found that in some cases leaving the "smoothness of the trend" parameter \sigma_trend unconstrained leads to a wiggly "trend" that captures all of the variability (with enough burn-in), and in the most recent version we use a half-Gaussian prior on \sigma_trend with variance 5e-4. The other parameters are left with improper uniform priors, and the AR correlation coefficient prior is updated to being uniform on [0, 1] rather than [-1, 1]— we agree that negative AR correlations are difficult to justify physically (although the strictly positive prior made little/no different in practice). In tests on simulated data we find that all hyper-parameters identify well under these priors - we have included new plots of the parameter posteriors in the appendix.

**General state space model approach**

I suggest that you describe the merge and trend analyses as a general hierarchical state space model. In both merging the data and in the DLM analysis you are dealing with a dynamical state space model. A general framework to describe the statistical model is by a hierarchical description, with a process model for the model state dynamics, a parameter model for model (nuisance) parameters and a data model for the likelihood. The Bayes formula would provide the posterior estimate from the individual conditional components as (see [1,2,3]):

[process, parameters|data] ∝ [data|process,parameters][process|parameters][parameters]

Filtering and smoothing algorithms can be used to estimate various marginal and conditional posterior distributions. The nuisance parameter could be integrated out by MCMC, for example.

For ozone data merge the process model includes the month-to-month variability and external events like volcanos, trends etc. The observation processes could describe the instrument effects. Lastly, there is the prior distributions for model parameters. The whole will in effect be a hierarchical Bayesian model to describe and estimate the state together with the parameters. This could provide a common framework for both merging and analysing.

We agree that the merge and trend analyses should really be done simultaneously in a single Bayesian hierarchical model (BHM). We have an on-going project where we are developing a sophisticated BHM for analyzing ozone data from scratch, going back to the original instrument records and modeling systematics explicitly rather than attempting to merge already-merged composites. However, this is well beyond the scope of the current paper, although it is a first step that resolves some of the key data-issues and is a coarse approximation to the full BHM approach.

A related issue that some readers have raised is concern over "using the data twice" — once to construct the transition prior (and uncertainty estimates) and once again in the main analysis (i.e. posterior sampling). Estimating the prior hyper-parameters and uncertainties a priori and fixing them can really be seen as approximation to the full BHM solution.

To cover these issues, we've added a section titled "BASIC as an approximation to a Bayesian hierarchical state-space model" where we briefly describe the full BHM approach and make explicit the fact that pre-computing the transition prior and uncertainties is an approximation to the BHM approach, which is good in the fortuitous case where those pre-computed quantities are strongly constrained by the data and do not strongly co-vary with the parameters of interest.

[1 ] L. M. Berliner. Physical-statistical modeling in geophysics. Journal of Geophysical Research: Atmospheres, 108(D24):8776, 2003. doi: 10.1029/2002JD002865.

[2 ] C. K. Wikle and M. B. Hooten. A general science-based framework for dynamical spatio-temporal models. TEST, 19(3):417–451, 2010. doi: 10.1007/s11749-010-0209-z.

[3 ] N. Cressie and C. K. Wikle. Statistics for Spatio-Temporal Data. Wiley, 2011.

**Other comments**

line 385, equation (2): I do not see how the parameters γ and β give rise to bimodality for an individual composite as the mean is the same $d^c_t$ for both modes. It probably will make the tails of the likelihood heavier than for a standard Gaussian likelihood.

The heavier tails of the likelihoods for individual data points leads to enhanced bi-modality when these likelihoods are multiplied together. See the new Figure. A3 and comment/response to reviewer 1 where the figure has been included there too.

line 460: The PF distribution is said not to be Gaussian but in DLM and MLR you need Gaussian uncertainty. Is this a problem for the trend analysis?

A "most principled" and optimal trend analysis will consider full non-Gaussian uncertainties. We are in the process of developing extensions to DLM that can deal with non-linear models and non-Gaussian likelihoods - however, this is well beyond the scope of this work. It's difficult to assess quantitatively to what extent the Gaussian assumption biases the trend analysis without knowing the "right answer" accounting for non-Gaussian errors. However, we feel that the impact of non-Gaussian errors is one of a large number of remaining deficiencies in trend analyses performed in the community, such as the linear-model assumption, fixed regressor phases etc. It is very likely not the biggest evil in this basket of remaining issues.

line 801: "using the same instrument dataset more than once". The transition prior is inferred from the same observations that are used in the model, so the data is used twice. Also, the uncertainty is inferred from the same data by SVD. Maybe this is ok here, but it violates the Bayesian assumptions.

See discussion above under "General state-space model approach" — pre-estimation of the transition prior and uncertainties can be thought of as an approximation to the full Bayesian hierarchical model. We leave the full hierarchical treatment to future work.

Can you elaborate more the claim that PF method can resolve the problems in data merging? Do you claim that PF is capable to extract the background truth behind different biased estimates. Or does it just make the "error bars" larger, so that the trend analysis is not affected by instrument artefacts?

The heavy-tailed Gaussian-mixture likelihood combined with the transition prior is able to identify where one or more datasets are biased, and result in a posterior whose mean is un/less-biased without necessarily ballooning the error bar. This can be seen from the fact that the product of Gaussian-mixture likelihoods can result in a multi-model joint-likelihood where the widths of the individual modes are not expanded as much as for a product of normal Gaussians. If the multiplication of the transition prior then excludes one of these modes, the resulting posterior effectively rejects the data in the excluded mode and what is left does not necessarily have an inflated uncertainty.

I agree that construction of a merged data set is of interest in itself. For trend analysis one could start from individual observations. You could discuss the possibility of a general data fusion approach that assimilates all the different composites or individual retrievals to a common time series model. You might still be able to use linear model, but with carefully designed (linear) observation operator, that would account for the instrument artefacts. Or use some non-linear generalization of DLM.

As discussed earlier, we completely agree that this is the way forward and have an exciting on-going project concerned with exactly this problem, but we feel it's beyond the scope of the current paper.

***Conclusion***

I can recommend the article to be published, if the author formulate the modelling approach for merging and uncertainty estimation a little more consistently, motivate the adequacy of the filter in the data merge and the use of SVD for the uncertainty variance components, and describe the MCMC results for DLM.

---

## Author Comment (AC2) · 28 Jul 2017

**Atmos. Chem. Phys. Discuss., doi:10.5194/acp-2017-142-RC2, 2017**

**General response**

We thank both referees for their thorough consideration and constructive feedback. As a result of the review process, we have made a significant effort to improve semantics regarding methods, models and algorithms. In the revised version of the manuscript we have replaced the particle filtering method by a Hamiltonian Monte Carlo (HMC) approach to sample the full posterior distribution, conditioned on the full data vector as is required, rather than just the data up to time t, as in the particle filter. HMC is well documented in the literature, and as such, the length section on the particle filter has been significantly reduced. We re-did the sampling algorithm from scratch, re-ran everything and re-made all relevant plots. In practice, we found that the new results are broadly similar to the particle filtering results and none of the key findings are changed. We now refer to the composite constructed using Gaussian-mixture likelihood and transition prior, with SVD uncertainty estimates, as the BAyeSian Integrated and Consolidated (BASIC) composite.

We note that there are some differences that you should be made aware of compared to the previous version. These include:
- the time-dependent error bars are much tighter, and much closer to Gaussian than before; this is good because the DLM analysis will better represent the data with Gaussian errors included;
- the problem we found (in only a few limited regions) following Pinatubo has gone and BASIC performs well during this period, given the data supplied to the process;
- there is a longer section in the appendix that encompasses requests for information on the Gaussian-mixture likelihood construction (in BASIC), and the parameter estimation (in the DLM);
- Northern and southern hemispheres in the profiles were actually the wrong way around; given the symmetry between hemispheres, the conclusions do not change.

We reply to all comments below, with referee comments in black, and our responses in blue.

**Daan Hubert (Referee)**
**Received and published on 20 April 2017**

2. Minor comments

• l.142, p.5: There is still a few % diurnal component between 1-5 hPa which will alias into the long-term trend for uncorrected measurements from instruments on drifting orbits. Please clarify that diurnal variations are not entirely avoided, only those with largest magnitude above the stratopause.

Added "; note, however, that some diurnal variability exists down to 5 hPa." at the end of the sentence.

• l.172-177, p.7: I doubt an uninformed reader will grasp the message in "[. . .] SBUV-MER considers only one data set at a time [. . .], while SBUV-MOD averages overlapping data to combine them [. . .]". A slightly more verbose description of the SBUV-MER merging approach may make the difference with that of SBUVMOD clearer.

This paragraph has been rewritten and now states: "The two SBUV-composites built in two different ways: SBUV-MER uses overlapping timeseries (shading in Fig. 1) to calculate offsets (calibration

biases) and differences in seasonal and diurnal variation, but only a single dataset is used without averaging overlapping periods; SBUV-MOD also accounts for offsets, but then overlapping data are averaged. SBUV-MOD relies on the instrument to instrument calibration done at the wavelength level within the version 8.6 algorithm for absolute calibration (i.e. no additional offsets are applied before averaging)."

• l.222-223, p.9: Not all occultation instruments retrieve O3 number density on altitude levels, only the UV-VIS instruments do so (SAGE-II). IR occultation missions (HALOE, ACE-FTS) retrieve O3 volume mixing ratio, some even on pressure levels. Please correct this statement.

This now reads as: "Occultation satellites measure ozone by looking at the disk of the rising or setting Sun though the atmosphere (SAGE-II uses the UV and visible, while e.g. HALOE and ACE-FTS use infra-red wavelengths); this makes their vertical profile resolution higher, but at the expense of only observing 15 profiles per day."

• l.352-362, p.13: What's the rationale for the factor 2 increase, and how sensitive are the Particle Filter results to this choice? Over what timescale is the uncertainty expanded? Just the month following the change or is it smeared out over a number of months?

The answer is not so simple, as it depends on the number of datasets available, how much in agreement they are, and how long the increased uncertainty is applied for; the decision is in part subjective. An example at 4.6 hPa and 0-10°N is provided below to help explain; it contains the four composites (colours), BASIC composite result (i.e. uncertainties increased by a factor of x2; black) and BASIC with uncertainties not increased (x1, grey), increased by a factor of x5 (black, dashed) and by x10 (grey, dotted); also shown is the delta-O3 relative to BASIC (middle plots) and the standard deviation (bottom plots) in BASIC (grey/black hues) along with the absolute difference between the BASIC(x2) and the composites (colours). If all four composites are available then the increased uncertainty will not have much effect when a single month is being considered (e.g. a change in instrument) as can be seen at months when vertical lines appear in the plots, but this is not the case if only two were present (not shown). Also, a discontinuity is not guaranteed for single instrument changes, though one is clear at the first red-vertical line in the right, middle plot - the presence of two other datasets here prohibits a jump forming in BASIC regardless of the increased uncertainty (factor) applied.

The effect of enhancing uncertainties is much clearer for periods when an extend enhanced-uncertainty is enforced (where the black filled rectangles appear in the upper two plots). It is clear in the left plots that *not* applying an enhancement to the uncertainty in SBUV-MER (yellow) leads to a rapid deviation following the Mt. Pinatubo eruption and a blowing up of the uncertainty for the period SBUV-MER is divergent from the SWOOSH/GOZCARDS group; we have been told (J. Wild, private communication) that SBUV should not be considered during this period, and the data in the SAGE-based composites were cleaned for artefacts related to high sulfate-levels. In the latter period (right plots), the uncertainty is increased for the pair of SBUV-composites because of the known drift. Not enforcing this introduces a small positive drift that is transferred to the composite from the drift in the SBUV-composites (i.e. it is not fully removed).

[Figure]

Increasing the uncertainty enhancement to a factor of five or ten, generally makes little change to the BASIC result from a factor of two, but there are some instances when it leads to strongly following GOZCARDS and SWOOSH (e.g. in the right plot), which may not be appropriate, but in general this last only a few months before returning to the x2 level. Thus, we remain conservative by applying a factor of x2 only, such that most of the large deviations are accounted for but without applying an enhancement that is so large (e.g., x10).

• l.368-369, p.13: How prohibitive is the assumption of uncorrelated measurement errors for the joint-likelihood function? The bottom row of Fig. 4 clearly demonstrates the correlation of the uncertainties in time and between composites.

The uncertainties derived from the SVDs are not true error estimates, but are an uncertainty related to the deviation of the composite from the pack. Thus, artefacts in a single composite are mainly encoded within the uncertainty of that single composite, while deviations in both composites within a pair will be encoded into all the composite uncertainties, meaning they are correlated to some degree, so it is fair to say that there is some correlation in the SVD uncertainties. However, this is different to the uncertainties that increase together in all composites as a result of increased uncertainty simply due to their seasonal dependence, or from the true instrument (shot) 'noise' which is certainly uncorrelated between instruments. When the same instruments are used in multiple composites, then the uncertainties are repeated in both composites (and therefore inflated overall) and should be enhanced to reflect their double use, though this is not straightforward to achieve – we have now made this clear in the manuscript.

To keep this fair in the PF composite, we restricted our analysis to two pairs based on approximately the same instruments to avoid it, but it may lead to slightly tighter confidence intervals in the PF composite than would reflect essentially two fully-independent datasets.

• l.377-379, p.14: Assuming that β = 10% of the observations need a blow-up of their uncertainty by γ = 100 is quite harsh. I would expect smaller values for β and especially for γ, whose effect would be to reduce the tails of likelihood. But perhaps your choice is more of a worst-case scenario? How sensitive is the Particle Filter outcome to the choice of γ and β?

We now include an additional figure (below) in the appendix with brief explanation in the caption and subsection (entitle "Effect of the Box-Tiao equation") linked to the main part of the manuscript discussion on this equation. The effect can be quite significant given the choice. In the Figure we show 25 plots (5 values of g=gamma combined with 5 values of b=beta). In this plot we imagine an idealized scenario of 4 data points at -1.5, -1, +1, and +1.5, all with an uncertainty of sigma=0.2.

It is clear that for either low values of g, and/or low values of b, we get the expected result assuming all data is independent (which is what the dotted line in all plots), but this is inadequate as the pdf (dotted line/black thick line) does not represent any of the data and is in a region of lowest probability. For large values of b AND g (top right) we end not believing any of the data points at all (i.e. we enhance sigma^2 by a factor of gamma^2) with any affect from the second (separation) term (1-b)*exp[…] killed off by b~1; clearly this state is also incorrect. As the aim is to essentially enhanced regions where data agree, and kill off outliers, the preferred region of interest is for intermediate values of b (0.1-0.9) and g>10. From this, we choose b=0.1 and g = 100 as this appears to reflect well the desired separation into a multimodal pdf that represents two independent sets of data (e.g. blue and red/yellow groups).

[Figure]

In terms of its effect on the particle filter timeseries, when combined with a prior expectation, this can lead to the expected timeseries following one pair after it has become clear that jump/offset has occurred, whereas low g or low b leads to getting an average of all the composites with a bias introduced by the prior.

• l.403-413, p.15: It should be mentioned here that no transitions were used when an instrument changed. This relevant information is now hidden in the caption of Fig. 5.

We have added "; data in a composite where instruments change in not included at this stage" in the text shortly after the first mention of this figure.

• l.440, p.15: I had to wait for 43 lines to find out how large N is. I would mention this already from the start and come back to its motivation at the end of the section.

Added: "we generate $N$ (= 10,000) particles".

• l.962, p.38: I don't see months 20-30 as a second exception, they are just a result of the offset of all four composites in month 50.

We have removed "except around months 20-30".

• l.973-974, p.39: This phrase confused me, do you mean that the vertical resolution of SBUV degrades at lower altitudes? Perhaps you canted to say "This difference in vertical resolution becomes more important at lower altitudes."?

Yes, and we have replaced that part of the sentence with your clearer formulation, thank you.

• l.441-442, p.17: You praised the benefit of using a non-Gaussian likelihood (sum of Gaussians) in Sect. 3.2, so it is confusing to read about Gaussian composite likelihoods here. My eye cannot distinguish the likelihoods in Fig. 6 from Gaussian distributions (which also touches on the topic in a previous comment about β and γ), the former should be more heavy-tailed. I would just drop the "as Gaussian distributions".

Done.

• l.45-462, p.17: I found these couple of phrases (ending with "Fig. 6c.") of little value for the paper, as they essentially give a technical explanation of the resampling procedure. Or did I miss something?

We have changed "In this way, particles with higher weights are resampled more frequently and thus the posterior distribution represents the prior multiplied by the likelihood, i.e. the posterior in Fig. 6c." to simply read "In this way, particles with higher weights are resampled more frequently."

• Fig. 6, p.18: You may want to point somewhere in the paper to the outlying GOZCARDS likelihood in panel (j) which has a clear impact on the 99% credible region. I found this a nice illustration of the multi-modal joint likelihood.

This is a nice suggestion, which we have incorporated into the text. We have added at the end of the description of this Figure: "It is worth pointing out that 99% credible region of the posterior in Fig. 6j clearly deviates from a Gaussian distribution, caused by the deviation of GOZCARDS from the group, and is a real-data example of the multi-modal joint likelihood formed from using Box-Tiao (equation 1)".

• l.536-539, p.20: Is the transition prior of PF(SAGE) bootstrapped from the transitions of the two SAGE-composites rather than from the four composites?

Yes. To make this clear we have added the bold text: "But, it is also possible to only use information from either the SBUV-pair (`PF(SBUV)') or SAGE-pair (`PF(SAGE)') of composites **(with SVD uncertainties constructed using only the respective pairs of data)**, …"

• l.553-554, p.20: This phrase is strange, perhaps part of it is missing. How can local time of equator crossings be near-polar to attain near-global coverage?

We agree this was not clear. We have added the bold text to clarify: "Ideally, the local time at equator crossings should be the same each orbit, and **the orbit should be** near-polar to attain near-global coverage."

• l.589, p.23: OSIRIS is a limb-viewing instrument, so should not be mentioned here.

We have removed this error.

• l.606, p.23 (and elsewhere): The notion of "trend" carries various meanings in the community. Personally, I preserve "trend" for any long-term component that can not be attributed to known atmospheric processes or to known measurement artefacts. I advocate the phrasing "drift" or "artificial trend" in the latter case, which is much less confusing than blending it in with actual geophysical signal.

We agree. In multiple places throughout the manuscript we have revised the use of trend and limited it to only referring to the long-term 'real' change in the background ozone not accounted for in the quasi-periodic regressors.

• Sect. 4.3: How did you go from the time series in 10 degree latitude zones to regression results over 30 degree wider latitude belts? Average the time series, then regress? Please explain this briefly in the manuscript.

To the end of the first paragraph of section 4.3, we added: "These integrated latitude bands were formed by averaging the area/latitude-weighted 10°, with the 30-40° band receiving half the weight of the equivalent full band; the resultant timeseries were then analysed." Interestingly, we found very similar results taking the approach of considering each band separately and then averaging the profiles and adding variance in quadrature.

• l.752-754, p.28: Do you (or Laine et al) have an explanation for this instability? If you don't, perhaps mention that this deserves further investigation. This feature is striking and should be better understood.

No, and this was of serious concern to us too. It is exactly why we mentioned in section A4 that "A more thorough assessment of the DLM with respect to MLR will be made in a forthcoming publication." It is something we are in the process of evaluating. As that sentence is tucked away in the appendix, we have added "and requires investigation in a future publication to understand." to the end of the sentence mentioning it.

• Fig. A2, p.34: Specify the latitude range unless the Figure is for the entire data set at 1 hPa.

The original figure was for 0-10N, but we have changed this to 20S-20N. Actually, SBUV observations are pressure independent. Also, the y-axis title was incorrect, as the square-root of the number of observations were shown, which has been corrected.

• l.894, p.34: Add units to "small drifts of 0.5%". I know many people refer to 0.5% per year (or 5% per decade) as small, but they actually mean small compared to the stability of the data records. It is definitely not small compared to the actual trend being targeted, so this is a very unhappy phrasing

in my opinion. Same comment for "[...] SAGE and HALOE agree to within 5% in terms [...]", what is the unit?

You are quite right. For the first we have added 'yr-1' as the units for the drift. We have removed the second part of the sentence: the five percent in terms of temporal stability came explicitly from Froidevaux et al., 2015, but we could not find this explicitly mentioned in the reference, so we have modified the text based on your recommendation in the next point, below. We have changed the text to read: "We note that small drifts of ~0.5%yr-1 do exist between HALOE, SAGE II and Aura/MLS (Nair et al., 2012; Kirgis et al., 2013), and Nazaryan and McCormick (2005) and Hubert et al. (2016) suggest that most the datasets used in GOZCARDS have good stability."

• l.895, p.35: Hubert et al. (2016) is the first official report of a significant drift of 5% per decade of HALOE relative to sonde and lidar. Previous studies are consistent with this negative drift, but the results were not significant. I suggest to nuance your statement slightly: "[...], Nazaryan and McCormick (2005) and Hubert et al. (2016) suggest that MOST datasets used in GOZCARDS have good stability."

See previous comment.

• Algorithm 1 (Step 3), p.36: See earlier comment, the composite likelihoods are not Gaussian according to Eq. 3.

We remove 'Gaussian' from this sentence.

• Algorithm 1 (Step 4), p.36: See earlier comment, isn't this just a technical description of implementation? I would summarise this to one phrase.

Yes, this is aimed at being a technical description so that others can code up this should they wish. We would prefer to keep it technical since this is the most complicated step to understand in the algorithm.

• l.950, p.37: The word "original" is somewhat ambiguous here; it could mean the real, observed time series or the fit (with/without Gaussian noise?) to that time series. I find the phrasing "undamaged" time series better here (also used as label in Fig. A4b).

We agree the terminology is clearer with 'undamaged' and have changed all occurrences related to this.

**3 Technical corrections**

• l.125, p.4: The Penckwitt et al. paper was (and will likely) not (be) published. Please double-check this with G. Bodeker, one of the authors. Alternatively, Tummon et al. (2015) probably remains the best reference for this data set, as it has a concise summary of the merging method, satellite instruments and data versions.

Changed to Tummon et al., 2015.

• l.181-182, p.8: Not sure where the "this" refers to in "[. . .]; this describes the updated version [. . .]".

Changed to "which is an" and made changes to grammar before and after; it refers to the SBUV-MER change to only use NOAA-9 between 1994 and 1997.

• l.207, p.8: Reference to Fig. 2b, should be to Fig. 2c.

Done

• Fig.3, p.9: It is hard to discern blue from black markers/lines in print. Perhaps this Figure will benefit from a deviation of the colour scheme used in the rest of the paper.

Done; colours changed to red/pink hue.

• l.233, p.10: Replace "SAGE-II-based instruments" by "SAGE-II".

Done

• l.283-286, p.11: The section references are incorrect.

Done

• l.378, p.14: Smaller values of β encode more faith in individual observations rather than less, no?

Yes, you are correct. This has been changed.

• l.390, p.15: "compositeS".

Done

• l.435, p.17: Remove "(Algorithm 1)" following "the preparation step".

Done

• l.672, p.25: "deseasonAlised".

Done

• l.730-731, p.27: A "negative decrease" is, strictly speaking, an increase. Could be replaced by "[...], and insignificant decreases at [...]".

Agreed; we replaced this with "and negative but insignificant trend at lower altitudes".

• l.873, p.34: The colon messes up the citation. Which of Frith and DeLand (perhaps both) recommends that NOAA-9 should not be used?

Done; specifically it was DeLand.

• l.878, p.34: Remove duplicate "to" in "[...] tending to to cancel [...]".

Done

• l.895, p.35: Hubert et al. (2015) became Hubert et al. (2016) in the meantime.

Done

• l.914, p.35: Add a reference to Algorithm 1 on the next page, so this section is not empty.

Done

• l.937, p.36: Typo in "sectioN".

Done

• Fig. A4, p.38: Fix the legend label for "DLM" in the bottom panel.

Done

• Fig. A5, p.39: Fix labels in caption, should be (a) and (b) instead of (b) and (c).

Done

• l.975, p.39: Add a space before reference to Bhartia et al.

Done

• l.1168-1174, p.45: Update reference to AMT version of manuscript.

Done

• l.975, p.39: Add a space before reference to Bhartia et al.

Done

---

## Author Response (AR2)

**Response to final comments on**
**"Reconciling differences in stratospheric ozone composites"**
**by William T. Ball et al.,**
**Atmos. Chem. Phys. Discuss., doi:10.5194/acp-2017-142-RC2, 2017**

**General response**

We thank Marko Laine for the time he has taken to consider the manuscript, once again and for a second time.

We reply to all comments below, with referee comments in black, and our responses in blue.

**Marko Laine (Referee)**
**Received and published on 21 August 2017**

The acronym BASIC does not stand out very well, so it might not be a good choice if the authors would like to advertise their approach and it to easily found by search engines.

While we appreciate the concern in finding the composite, we have decided to keep the acronym, which we find to be quite descriptive. We also provide a reference and URL to the composite, which should make it easier to locate.

You only mention MCMC once in line 293. You could note that the HMC algorithm used in data merge is an MCMC algorithm, too, as well as that the posterior analysis for the variance parameters in the DLM model is done by MCMC sampling (if this is the case).

We have included more specific references to the fact that HMC is an MCMC algorithm.

In the subsection 4.2 "Example results of the BASIC approach", you could more clearly state that you are dealing with synthetic data sets. The title could be e.g. "Testing BASIC with synthetic data".

We have done this, and added replaced 'tests' with 'synthetic tests' in the first paragraph.

The marginal posterior densities in Fig. A4-A7 could be smoothed a little as the wiggles are probably caused by small sample size and the choice of kernel density estimator bandwidth parameter and are not showing real features in the posteriors.

We have updated the posteriors to make them smoother, i.e. by decreasing the number of histogram bins.